# Strategy Executability in Mathematical Reasoning: Leveraging Human–Model Differences for Effective Guidance

**Weida Liang** [1]  **Yiyou Sun** [2]  **Shuyuan Nan** [1]  **Chuang Li** [1]  **Dawn Song** [2]  **Kenji Kawaguchi** [1]

## Abstract

Example-based guidance is widely used to improve mathematical reasoning at inference time, yet its effectiveness is highly unstable across problems and models—even when the guidance is correct and problem-relevant. We show that this instability arises from a previously underexplored gap between *strategy usage*—whether a reasoning strategy appears in successful solutions—and *strategy executability*—whether the strategy remains effective when instantiated as guidance for a target model. Through a controlled analysis of paired human-written and model-generated solutions, we identify a systematic dissociation between usage and executability: human- and model-derived strategies differ in structured, domain-dependent ways, leading to complementary strengths and consistent source-dependent reversals under guidance. Building on this diagnosis, we propose *Selective Strategy Retrieval* (SSR), a test-time framework that explicitly models executability by selectively retrieving and combining strategies using empirical, multi-route, source-aware signals. Across multiple mathematical reasoning benchmarks, SSR yields reliable and consistent improvements over direct solving, in-context learning, and single-source guidance, improving accuracy by up to $+13$ points on AIME25 and $+5$ points on Apex for compact reasoning models. Code and benchmark are publicly available at: https://github.com/lwd17/strategy-execute-pipeline.

---

[1]National University of Singapore [2]University of California, Berkeley. Correspondence to: Kenji Kawaguchi <kenji@comp.nus.edu.sg>.

*Proceedings of the $43^{rd}$ International Conference on Machine Learning*, Seoul, South Korea. PMLR 306, 2026. Copyright 2026 by the author(s).

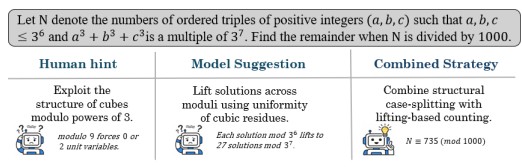

Let N denote the numbers of ordered triples of positive integers $(a, b, c)$ such that $a, b, c \leq 3^6$ and $a^3 + b^3 + c^3$ is a multiple of $3^7$. Find the remainder when N is divided by 1000.

| Human hint | Model Suggestion | Combined Strategy |
|---|---|---|
| Exploit the structure of cubes modulo powers of 3. | Lift solutions across moduli using uniformity of cubic residues. | Combine structural case-splitting with lifting-based counting. |
| *modulo 9 forces 0 or 2 unit variables.* | *Each solution mod $3^6$ lifts to 27 solutions mod $3^7$.* | $N \equiv 735 \pmod{1000}$ |

*Figure 1.* An example illustrating that strategies which appear valid in isolation may fail when transferred as guidance. In this AIME-level problem, a human-derived structural strategy and a model-derived procedural strategy are each insufficient on their own, while selectively combining them enables successful execution.

## 1. Introduction

Large language models (LLMs) have demonstrated strong performance on mathematical reasoning tasks, particularly when augmented with inference-time guidance such as worked examples, concise hints, or high-level reasoning suggestions (Wei et al., 2022; Lewkowycz et al., 2022; Kojima et al., 2022; Achiam et al., 2023; Brown et al., 2020; Yao et al., 2022). When effective, such guidance does more than add information: it steers the model toward a particular solution strategy and shapes the sequence of reasoning steps it attempts to execute.

Despite these gains, guidance-based reasoning remains strikingly unreliable. Across models and benchmarks, even guidance that is demonstrably correct, problem-relevant, and extracted from successful solutions often fails to help—and can sometimes degrade—performance (Madsen et al., 2024; Guo et al., 2025). These failures recur across domains and model families and cannot be explained by deficiencies in guidance quality or semantic relevance alone, pointing to a deeper limitation in how guidance is currently understood.

A central assumption underlying existing approaches is that a reasoning strategy observed in a successful solution can be reliably carried out when transferred as explicit guidance to a target model. In practice, example-based guidance primarily conveys *reasoning strategies*: high-level decisions about problem decomposition, representation, and solution structure (Simon & Newell, 1971; Chi, 2006). Human-authored solutions, in particular, often emphasize conceptual insight and global structure (Larkin et al., 1980; Polya,

1957). These human strategies are typically concise, abstract, and under-specified, relying on implicit reasoning steps that may not align with the operational strengths of a target model. As a result, the mere presence of a strategy in a correct solution does not ensure that the target model can effectively use it when prompted.

This gap highlights a distinction that has received little explicit attention: the difference between whether a strategy appears in a successful solution and whether it *remains effective as guidance* for a target model. We refer to the latter as *strategy executability*. Importantly, executability is assessed operationally—by whether providing the strategy as guidance under fixed prompting and decoding conditions increases the target model's likelihood of producing a correct solution, without requiring faithful step-by-step imitation of the strategy. This perspective leads to a natural question:

*Under what conditions does a reasoning strategy remain executable when transferred as guidance to a target model?*

To address this question, we adopt a strategy-level diagnostic perspective on mathematical reasoning. Rather than treating solutions as reasoning traces, we represent each solution as a composition of high-level strategies. This abstraction disentangles two notions often conflated in prior work: *strategy usage*, which captures how frequently a strategy appears in successful solutions, and *strategy executability*, which reflects whether the strategy remains effective when instantiated as guidance for a given model.

Using this framework, we analyze paired human-written and model-generated solutions to the same mathematical problems. Although both sources often arrive at correct answers, they do so using systematically different strategies: human solutions rely more on structural insights and conceptual decompositions, whereas model-generated solutions favor procedural and algebraic transformations (Trinh et al., 2024; Mahdavi et al., 2025). As we show, these differences have concrete consequences for guidance, shaping which strategies remain executable when transferred.

Figure 1 illustrates this phenomenon. When transferred individually under identical prompting conditions, neither strategy succeeds; only their selective combination yields an executable reasoning path. This highlights our central insight: effective guidance depends not on strategy presence alone, but on executability for the target model.

Motivated by this diagnosis, we propose **Selective Strategy Retrieval (SSR)**, a lightweight inference-time framework that explicitly models strategy executability. SSR selectively retrieves strategies from human-written and model-generated solutions based on empirical executability signals. It operates purely at test time and requires no modification to the underlying model, training data, or decoding procedure.

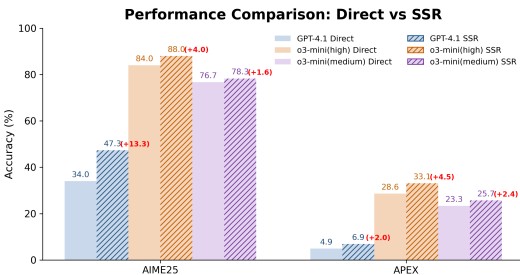

*Figure 2.* Performance gains from Selective Strategy Retrieval (SSR) on closed-source reasoning models (GPT-4.1 and o3-mini), measured by pass@1 and averaged over five runs.

Empirically, SSR yields consistent improvements across open-source and closed-source reasoning models. On closed-source models, SSR improves accuracy over direct prompting by approximately $+\mathbf{4} \sim \mathbf{13}$ points on AIME25 and $+\mathbf{2} \sim \mathbf{5}$ points on Apex for GPT-4.1 and o3-mini (Figure 2), demonstrating that explicitly modeling strategy executability is key to robust reasoning gains.

Our contributions are summarized as follows:

- We identify a systematic dissociation between strategy usage and executability in mathematical reasoning.

- To enable controlled analysis of this dissociation, we construct HM-ReasoningBench, a paired dataset of competition-level problems with human-written and model-generated solutions.

- Building on this diagnosis, we propose Selective Strategy Retrieval (SSR), a test-time framework that operationalizes strategy executability through selective strategy combination.

- We demonstrate that explicitly modeling strategy executability—rather than strategy prevalence or semantic relevance—leads to robust improvements across multiple mathematical benchmarks.

**Conflict of Interest Disclosure.** The authors declare no financial or substantive conflicts of interest related to this work.

## 2. Related Works

**Example-Based Reasoning Guidance.** Inference-time guidance, such as worked examples, reasoning traces, or high-level hints, is widely used to improve reasoning in large language models (Brown et al., 2020; Wei et al., 2022; Yao et al., 2022; Kojima et al., 2022; Liu et al., 2022; Zhou et al., 2022; Rubin et al., 2022; Shum et al., 2023; Wu et al.,

2023; Fernando et al., 2023; Diao et al., 2024; Zhang et al., 2025; Cao et al., 2025). While effective in some cases, prior work largely assumes that guidance is transferable across models and contexts, typically selecting examples based on semantic similarity or correctness (Zelikman et al., 2022; Yao et al., 2023). Recent studies show that additional guidance can be unreliable and may even degrade performance for certain models or tasks (Madaan et al., 2023; Guo et al., 2025), suggesting that the key challenge lies in whether guidance is executable by the target model.

**Reasoning Traces and Strategy Abstraction.** A large body of work represents solutions as step-by-step reasoning traces for supervision, explanation, or iterative refinement (Cobbe et al., 2021; Wang et al., 2022; Zelikman et al., 2022; Chowdhury & Caragea, 2025; Mukherjee et al., 2025; Jiang et al., 2025). However, recent work questions the faithfulness of such traces, noting that they may be post hoc or weakly coupled with model decision-making (Creswell & Shanahan, 2022; Xu et al., 2024; Wu et al., 2025; Munkhbat et al., 2025). This has motivated abstractions toward higher-level reasoning structure (Yu et al., 2025), as well as process-level methods such as process reward models (Hu et al., 2025; Younsi et al., 2025), which typically operate during training or decoding.

Related approaches introduce explicit strategy- or plan-level control, such as routing problems to strategies or selecting plans prior to generation (Xu et al., 2025; Qi et al., 2025), but do not analyze whether such strategies remain executable across models or contexts.

In contrast, we treat reasoning strategies as analytical objects. We abstract strategies from human-written and model-generated solutions and study *strategy executability*—whether a strategy that appears in a given solution can be operationalized as guidance—revealing a systematic gap between strategy prevalence and effectiveness. This perspective motivates selective strategy retrieval based on empirical executability signals and human–model differences.

## 3. Strategy-Level Differences Between Human and Model Solutions

Before assessing whether a reasoning strategy can serve as effective guidance, we examine how strategies are employed by different solvers. Although human-written and model-generated solutions often reach correct answers on the same problems, they do so through systematically different strategic choices. This section provides a strategy-level analysis of these differences across problem domains and establishes the empirical foundation for the executability study in Section 3.4.

### 3.1. Dataset and Paired Solution Setting

We conduct our analysis on **HM-ReasoningBench**, which contains 4,895 challenging mathematical problems, each paired with a human-written solution and a model-generated solution. Problems are drawn from Omni-Math (Gao et al., 2024) and HARP (Yue et al., 2024), with HARP restricted to difficulty level $\geq 6$. The dataset spans algebra, geometry, number theory, combinatorics, and mixed-topic problems; additional statistics are reported in Appendix A.1.

### 3.2. Strategy Abstraction

To enable strategy-level comparison, we represent each solution as a small set of high-level reasoning strategies. Each solution is associated with multiple strategies (typically 3–5), reflecting the compositional nature of non-trivial mathematical reasoning. Strategies are treated as unordered, non-exclusive attributes of a solution, and the analysis in this section concerns which strategies appear rather than how they are executed.

Strategies are extracted using a prompting pipeline designed to identify *transferable reasoning patterns* that generalize beyond individual problems. Extracted strategies are mapped via rule-based matching to a predefined library of 30 canonical strategy templates, defined based on standard competition guidebooks and canonical treatments of mathematical problem solving (Polya, 1957; Engel, 1998; Zeitz, 2016). Full prompt details and strategy category definitions are provided in Appendix B.1 and Appendix A.2.

To ensure comparability across problems, we apply per-problem normalization when aggregating statistics.

### 3.3. Strategy Usage Differences

We first compare strategy usage between human-written and model-generated solutions, both in aggregate and conditioned on problem domain.

**Global preferences.** Aggregated across all problems, human and model solutions exhibit clear but moderate differences in their overall strategy distributions. As shown in Figure 3, human-written solutions place greater emphasis on geometry- and structure-oriented strategies, including auxiliary constructions, symmetry, angle chasing, and invariant-based reasoning. Model-generated solutions, in contrast, rely more heavily on algebraic manipulations, coordinate formulations, and equation-driven transformations.

These trends align with established observations in mathematical problem solving: expert humans favor relational and structural abstractions, whereas contemporary reasoning models more often adopt procedural strategies that decompose problems into explicit symbolic operations (Ruis et al.; Trinh et al., 2024).

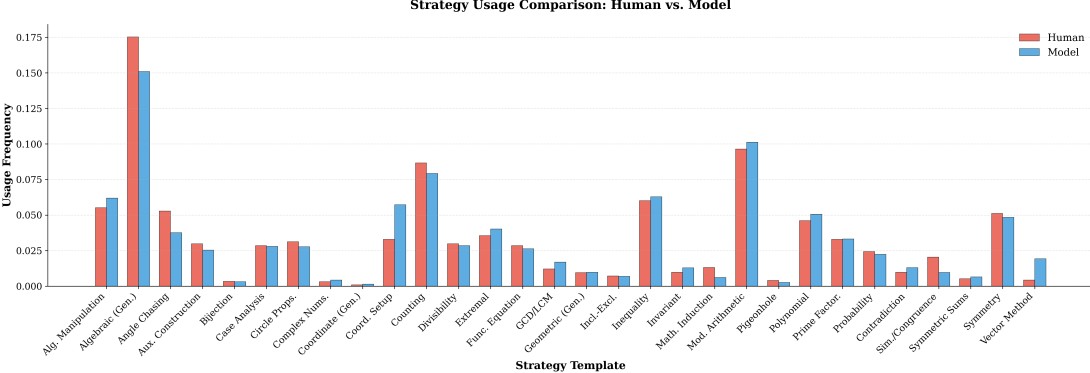

*Figure 3.* Normalized strategy usage in human-written and model-generated solutions, aggregated across problems with per-problem normalization. For each problem, strategies contribute equally, ensuring that multi-strategy solutions do not dominate the statistics.

**Domain-conditioned divergence.** Conditioning on problem subject reveals substantially sharper differences. Geometry exhibits the largest divergence: human solutions strongly favor construction- and relation-driven strategies, while model solutions disproportionately adopt coordinate-based reductions (Figure 4(a)). In contrast, algebra and number theory show much closer alignment in strategy usage, likely reflecting their more uniformly symbolic structure. Additional examples are provided in Appendix C.

These usage results should be interpreted as *descriptive* evidence of structured source differences rather than as direct evidence of guidance quality. They motivate the executability analysis below: if human-written and model-generated solutions organize reasoning differently, then the strategies they provide as guidance may also transfer differently.

> **Takeaway**
>
> Human-written and model-generated solutions differ systematically in the strategies they employ, with divergences that are strongly structured by problem domain. These usage patterns characterize how solutions are constructed, but do not indicate whether a strategy is reliable when transferred as guidance.

### 3.4. Strategy-Guided Accuracy Divergence

We now evaluate *strategy executability*: whether an individual strategy extracted from a solution can be operationalized by a target model when provided as explicit guidance.

**Setup.** We evaluate two compact reasoning models, **Qwen3-8B** and **DeepSeek-R1-Distill-Qwen-7B**. Each solution typically contains multiple extracted strategies. Rather than selecting a representative or random strategy, we treat each extracted strategy as an independent evaluation unit. For a given problem–strategy pair, the strategy is provided alone

as guidance under a fixed prompting and decoding protocol, and effectiveness is measured by final answer correctness. Results are aggregated at the strategy level; unless otherwise stated, each strategy is evaluated with multiple decoding trials and averaged to account for stochasticity. Both models exhibit consistent qualitative trends, so results are aggregated.

**Results.** Figure 4(b) reveals a clear dissociation between strategy usage and strategy executability. Strategies that are frequently used by a source solver do not necessarily yield higher accuracy when transferred as guidance.

Procedural strategies—such as *case analysis* and *coordinate setups*—are often more executable when sourced from model-generated solutions, particularly in geometry and mixed-topic problems. Conversely, structurally grounded strategies—such as *similarity/congruence* and *prime factorization*—transfer more reliably when derived from human solutions, despite being less prevalent in model usage.

One useful interpretation is that executability depends on two task properties: *abstraction leverage* (whether a high-level reformulation immediately reduces the problem) and *execution load* (whether success requires many explicit intermediate operations). Structural guidance tends to help more when abstraction leverage is high, whereas procedural guidance becomes more reliable when execution load dominates. We return to this task-conditional pattern in Section 5.5 and Appendix D.6.

## 4. Selective Strategy Retrieval

The analysis in Section 3.4 shows that naive strategy reuse fails for systematic reasons. In particular, using strategy frequency as a proxy for executability is unreliable; semantic relevance alone does not ensure operational alignment; and committing to a single strategy source ignores strong,

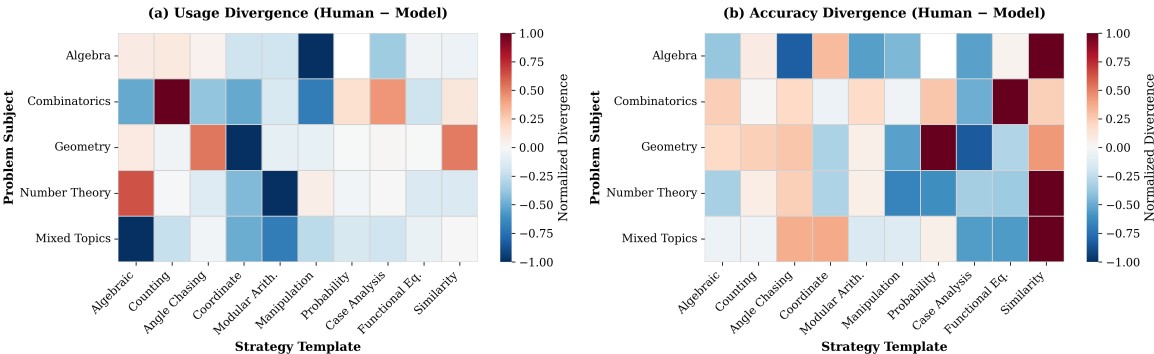

*Figure 4.* Strategy-level divergence between human-written and model-generated solutions. (a) Normalized differences in strategy usage. (b) Normalized differences in strategy-guided accuracy.

domain-dependent reversals. Together, these observations rule out retrieval schemes based solely on usage statistics, surface similarity, or uniform source preference. We therefore treat *strategy executability* as the primary criterion for strategy selection.

We introduce **Selective Strategy Retrieval (SSR)**, a test-time framework that selects strategies using source-dependent and context-conditioned executability signals and provides up to five strategies as guidance per problem.

### 4.1. Strategy Knowledge Graph

Executability is inherently relational: whether a strategy is executable depends on its interaction with problem structure, reasoning category, and source. We therefore organize reasoning knowledge from HM-ReasoningBench into a heterogeneous graph $\mathcal{G} = (V, E)$ with nodes corresponding to **problems** $V_p$, **strategies** $V_s$, and **categories** $V_c$. Edges encode observed problem–strategy usage and category membership. This representation allows executability signals to propagate across related problems while preserving the category-level regularities identified in Section 3.

**Source-aware retention.** Because executability depends strongly on strategy source, SSR does not treat all strategies within a category as equally reliable. For each category and strategy type, we preferentially retain strategies from the source (human or model) that exhibits higher empirical executability under guidance. This design retains complementary strategies when coverage is sparse.

**Graph representation learning.** To encode executability patterns, we learn structure-aware node embeddings over $\mathcal{G}$ using a heterogeneous graph neural network with transformer-style message passing. The model is trained with a contrastive objective that separates successful from unsuccessful problem–strategy pairings. The learned embeddings encode empirical signals of strategy executability

(Appendix A.4), which serve as structural features for downstream executability prediction, as described in Section 4.4.

### 4.2. Problem Representation

At test time, executability must be inferred for a new problem $x$ without direct supervision. We first embed $x$ using a pretrained sentence encoder to identify a neighborhood $\mathcal{N}(x)$ of semantically related training problems.

Rather than retrieving strategies by surface similarity, SSR aggregates the graph embeddings of problems in $\mathcal{N}(x)$ to construct a transferred representation $h_x$. Neighbors are weighted by semantic similarity via a temperature-scaled softmax, allowing relevant contexts to dominate. This representation provides a structure-aware abstraction of $x$, enabling retrieval based on learned executability patterns rather than semantic overlap alone.

### 4.3. Multi-Route Strategy Retrieval

No single notion of relevance reliably predicts executability. SSR therefore retrieves candidate strategies through three complementary routes, whose union forms the candidate set $\mathcal{S}(x)$ (Appendix A.3).

**Route A: Category-conditioned retrieval.** SSR retrieves strategies retained for categories compatible with $h_x$, capturing coarse-grained but robust executability signals that generalize across problems.

**Route B: Problem-transfer retrieval.** To capture fine-grained, context-dependent executability, SSR retrieves strategies that were empirically effective when guiding solutions to problems in $\mathcal{N}(x)$.

**Route C: Semantic fallback retrieval.** When executability evidence is sparse, SSR retrieves a small number of semantically similar strategies as fallback, ensuring coverage without assuming executability from similarity.

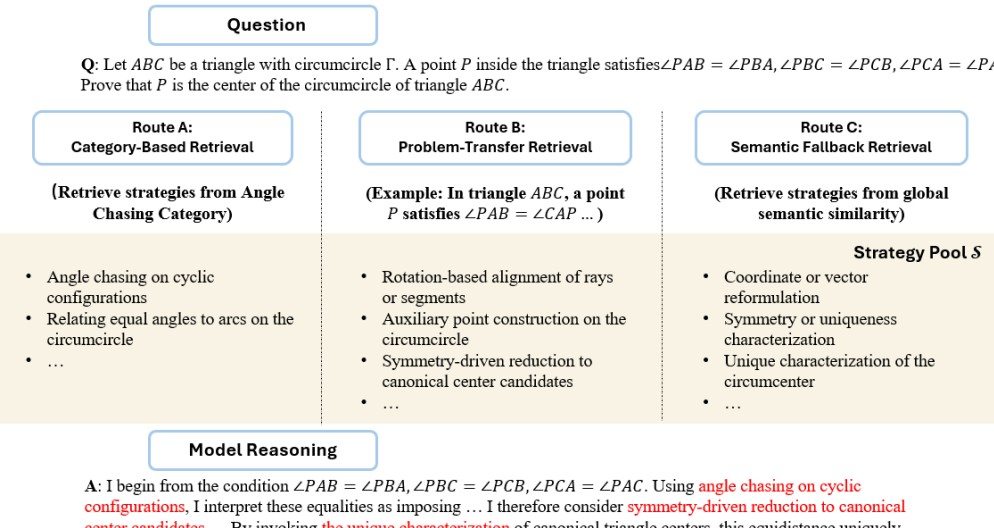

*Figure 5.* Multi-route strategy retrieval in Selective Strategy Retrieval (SSR). Complementary retrieval routes capture category-level regularities, problem-specific transfer, and semantic coverage, forming the candidate set $\mathcal{S}(x)$.

## 4.4. Modeling Strategy Executability

The multi-route retrieval step produces a diverse but over-complete candidate set $\mathcal{S}(x)$, within which only a subset of strategies are expected to be executable for the target model. Given a problem $x$ and a candidate strategy $s \in \mathcal{S}(x)$, SSR aims to estimate the utility of providing $s$ as inference-time guidance to a target reasoning model. We formalize it as a model-relative, protocol-relative quantity:

$$U(s \mid x; m, \pi) \;=\; \mathbb{P}\big(\text{success} = 1 \mid x, s, m, \pi\big), \quad (1)$$

where $m$ denotes the target model and $\pi$ denotes a fixed prompting and decoding protocol (including prompt template, temperature, and context budget). Intuitively, $U(s \mid x; m, \pi)$ captures the probability that providing strategy $s$ as guidance enables model $m$ to produce a correct solution for problem $x$ under controlled inference conditions. Success refers to pass@1 unless otherwise stated.

**Empirical supervision.** The executability utility in Eq. (1) is not directly observable. To obtain supervision, we evaluate strategy-guided execution outcomes on a training split of HM-ReasoningBench. For each problem–strategy pair $(x, s)$, we run the target model $m$ under protocol $\pi$ for $T$ independent decoding trials and record binary outcomes $y_{x,s,t} \in \{0, 1\}$ indicating whether the final answer is correct. We treat these outcomes as Bernoulli samples from an underlying success probability $p_{x,s}$ and estimate a posterior mean executability score using a Beta–Binomial model:

$$\tilde{U}(s \mid x) \;=\; \mathbb{E}[p_{x,s} \mid y_{x,s,1:T}] \;=\; \frac{\alpha + \sum_{t=1}^{T} y_{x,s,t}}{\alpha + \beta + T}, \quad (2)$$

with a weakly informative prior $(\alpha, \beta)$. In all main experiments, we use the symmetric choice $\alpha = \beta = 1$; sensitivity to other weak priors is reported in Appendix D.3. This formulation explicitly accounts for decoding stochasticity and yields a calibrated estimate of strategy executability.

**Executability predictor.** To generalize beyond observed pairs and enable efficient ranking at test time, we learn a parametric estimator $\hat{U}_\theta(s \mid x)$ that predicts executability from problem–strategy features. We construct a feature representation $\phi(x, s)$ that aggregates complementary signals, including: (i) semantic alignment between $x$ and $s$, (ii) structural proximity derived from the strategy knowledge graph, and (iii) route- and source-specific indicators reflecting how $s$ was retrieved. The executability predictor is defined as

$$\hat{U}_\theta(s \mid x) \;=\; \sigma\big(\theta^\top \phi(x, s)\big), \quad (3)$$

where $\sigma$ is the logistic function.

We train $\hat{U}_\theta$ using trial-level supervision by minimizing the negative log-likelihood of observed outcomes:

$$\mathcal{L}(\theta) = -\sum_{(x,s)} \sum_{t=1}^{T} \Big[ y_{x,s,t} \log \hat{U}_\theta(s \mid x) \\ + (1 - y_{x,s,t}) \log\big(1 - \hat{U}_\theta(s \mid x)\big) \Big]. \quad (4)$$

with $\ell_2$ regularization on $\theta$. This objective encourages $\hat{U}_\theta$ to approximate the true executability probability in Eq. (1).

To avoid notation ambiguity, we use three distinct quantities throughout: $y_{x,s,t}$ denotes a trial-level execution outcome,

$\tilde{U}(s \mid x)$ denotes the aggregated posterior executability target, and $\hat{U}_\theta(s \mid x)$ denotes the predictor output. The graph encoder and the executability predictor are also trained in two distinct stages: the graph model uses thresholded $\tilde{U}$ only to shape structural representations, while the final predictor is fit with trial-level outcomes. Appendix A.4 and Appendix D.5 provide the corresponding implementation details and ablations.

The role of the graph model is thus to provide structure-aware representations, while executability estimation and ranking are handled by a separate supervised predictor. **Calibration and ranking.** Because $\hat{U}_\theta$ is used for cross-route and cross-source comparison, we apply temperature scaling on a held-out validation set to calibrate predicted probabilities. At inference time, SSR ranks candidate strategies for problem $x$ by their calibrated utility scores and selects a small subset with highest estimated executability.

### 4.5. Using Strategies as Guidance

SSR outputs a small set of abstract strategy hints describing general reasoning approaches rather than concrete solution steps. At the strategy level, SSR preserves flexibility while aligning guidance with the operational strengths of the target model. The prompting format is shown in Appendix B.3.

## 5. Experiments

We evaluate whether executability-aware strategy selection, implemented by Selective Strategy Retrieval (SSR), reliably improves mathematical reasoning. Our experiments address three questions: (i) does SSR consistently improve accuracy across datasets and models, (ii) are its key components necessary, and (iii) when and why human-derived strategies are most effective.

### 5.1. Experimental Setup

**Datasets.** We evaluate on three benchmarks spanning paired analysis, competition-style reasoning, and extreme difficulty. **HM-ReasoningBench** is used to construct the strategy knowledge graph and is evaluated on a held-out test split. **AIME 2025** (Mathematical Association of America, 2025) contains competition-level problems requiring multi-step symbolic reasoning. **MathArena Apex** (Balunović et al., 2025) consists of highly challenging final-answer problems on which even strong models have low success rates, serving as a stress test for compositional reasoning.

**Models.** We evaluate **Qwen3-8B**, **Qwen3-14B**, and **DeepSeek-R1-Distill-Qwen-7B**. All models use the same configuration (max context 32,768; temperature 0.7).

**Metric and verification.** We report exact-match accuracy. For proof-oriented problems, we use GPT-5.1 to verify mathematical equivalence between model outputs and references; the verification prompt is provided in Appendix B.4.

**Reproducibility.** All reported results are averaged over 5 independent runs with different random seeds. We report mean accuracy throughout.

### 5.2. Baselines

All methods share the same prompting format and differ only in how guidance is sourced. **Direct Solving (DS)** solves the problem without external guidance; **In-Context Learning (ICL)** provides one worked example (more examples did not help and sometimes degraded performance); **Human-Only Guidance (H)** uses strategy hints extracted from human solutions; and **Model-Only Guidance (M)** uses strategy hints extracted from model solutions. In our implementation, the worked example used by ICL is itself selected by semantic retrieval; additional retrieval-based comparisons are reported in Appendix D.2.

We also compare against stronger inference-time baselines, including Self-Consistency (SC), Least-to-Most Prompting (L2M), and Tree-of-Thoughts (ToT), which allocate additional test-time computation through sampling or search.

### 5.3. Main Results

Table 1 reports accuracy across datasets. Three consistent patterns emerge. First, strategy guidance improves over DS in all settings, indicating that abstract strategy hints are generally usable by compact reasoning models. Second, **SSR consistently achieves the best performance**, outperforming both single-source guidance (H/M) and ICL, which rules out gains from merely adding more context. Third, SSR's relative advantage increases with benchmark difficulty, consistent with our executability analysis. Comparisons with stronger inference-time baselines, including self-consistency, least-to-most prompting, and Tree-of-Thoughts, are reported in Appendix D.1. **Notably, SSR achieves these gains using a single guided generation per problem, whereas these baselines allocate substantially more test-time computation.**

On HM-REASONINGBENCH and AIME25, both H and M improve over DS, while SSR yields further gains (e.g., Qwen3-8B: $63.80 \rightarrow 66.00/65.40 \rightarrow 68.60$ on the former, and $67.33 \rightarrow 70.67/69.33 \rightarrow 74.00$ on the latter). On the hardest benchmark APEX, SSR's gains are largest (e.g., Qwen3-8B: $8.16 \rightarrow 13.06$), reflecting the amplified impact of executability mismatches in long-horizon problems.

Across datasets, H slightly outperforms M on average. **SSR consistently improves over both** by selecting and combining strategies in a context- and source-aware manner. Qualitative examples illustrating how SSR yields more coherent reasoning trajectories are provided in Appendix D.9.

*Table 1.* Accuracy (%) comparison between Selective Strategy Retrieval (SSR), single-source guidance (H/M), in-context learning (ICL), and direct solving (DS) across three benchmarks. Best results are shown in bold.

| | HM-ReasoningBench | | | | | AIME25 | | | | | Apex | | | | |
|---|---|---|---|---|---|---|---|---|---|---|---|---|---|---|---|
| Model | DS | ICL | H | M | SSR | DS | ICL | H | M | SSR | DS | ICL | H | M | SSR |
| Qwen3-8B | 63.80 | 64.20 | 66.00 | 65.40 | **68.60** | 67.33 | 68.00 | 70.67 | 69.33 | **74.00** | 8.16 | 8.16 | 8.57 | 8.98 | **13.06** |
| Qwen3-14B | 67.40 | 67.60 | 69.40 | 68.20 | **70.20** | 70.66 | 70.33 | 72.67 | 72.00 | **74.67** | 11.02 | 11.43 | 13.47 | 12.65 | **14.69** |
| R1-Distill-7B | 49.20 | 49.80 | 51.00 | 50.80 | **52.40** | 42.00 | 48.00 | 51.33 | 52.00 | **53.13** | 4.08 | 4.90 | 6.53 | 6.12 | **7.75** |

*Table 2.* Ablation study of retrieval routes in Selective Strategy Retrieval (SSR) across three benchmarks. Best result is shown in bold.

| | HM-ReasoningBench | | | | AIME25 | | | | Apex | | | |
|---|---|---|---|---|---|---|---|---|---|---|---|---|
| Model | SSR | w/o Cat | w/o Tran | w/o Sem | SSR | w/o Cat | w/o Tran | w/o Sem | SSR | w/o Cat | w/o Tran | w/o Sem |
| Qwen3-8B | **68.60** | 66.00 | 63.60 | 67.20 | **74.00** | 70.67 | 67.33 | 72.67 | **13.06** | 12.24 | 11.02 | 13.06 |
| Qwen3-14B | **70.20** | 67.60 | 65.00 | 69.00 | **74.67** | 71.33 | 68.67 | 73.33 | **14.69** | 13.06 | 11.84 | 13.88 |
| R1-Distill-7B | **52.40** | 50.60 | 48.80 | 51.20 | **53.13** | 52.00 | 50.67 | 51.33 | **7.75** | 6.94 | 6.12 | 7.35 |

*Table 3.* End-to-end retrieval baselines on AIME25 (accuracy %).

| Method | Qwen3-8B | Qwen3-14B | R1-Distill-7B |
|---|---|---|---|
| BM25 retrieval | 67.33 | 69.00 | 47.33 |
| Dense retrieval | 68.00 | 70.33 | 48.00 |
| MMR semantic reranking | 68.67 | 71.00 | 49.33 |
| **SSR** | **74.00** | **74.67** | **53.13** |

To further isolate whether the gains come from executability-aware selection rather than retrieval alone, Table 3 compares SSR against end-to-end BM25, dense retrieval, and MMR-style semantic reranking baselines under the same prompt budget and decoding protocol.

SSR remains best across all three models, showing that the gains are not explained by semantic retrieval quality alone.

## 5.4. Ablation: Is Multi-Route Retrieval Necessary?

SSR constructs candidates via three retrieval routes corresponding to distinct executability signals. We ablate each route in turn while keeping all other components fixed: Route A (category-conditioned), Route B (problem-transfer), and Route C (semantic fallback).

As shown in Table 2, removing any route consistently degrades performance, indicating that no single signal is sufficient. Removing Route B causes the largest drop (e.g., Qwen3-14B on APEX: $14.69 \rightarrow 11.84$), highlighting the importance of fine-grained, context-dependent transfer. Removing Route A also leads to clear declines (e.g., Qwen3-8B on AIME25: $74.00 \rightarrow 70.67$), while removing Route C yields smaller but consistent drops, reflecting its role in maintaining coverage when executability evidence is sparse.

## 5.5. Analysis: When Does Human Guidance Help Most?

Our strategy-level diagnosis predicts that human-derived guidance helps most when failures are driven by missing *global structure* (e.g., absent decomposition, constraints, or case splits) rather than local symbolic slips. We test this prediction using topic-level and failure-mode analyses.

**Topic-level.** Figure 6 reports gains over DS on HM-REASONINGBENCH (Qwen3-14B). Human guidance yields the largest gains in geometry and combinatorics, while model guidance is weaker and can degrade performance. In algebra and number theory, source effects are smaller and both H and M provide modest gains. Across topics, **SSR matches or exceeds the stronger source**, confirming that source effectiveness is context-dependent.

This pattern can be understood through two complementary task properties. When the main bottleneck is finding the right *high-level reformulation*, human-derived structural guidance is often more executable. When success instead depends on carrying out a long sequence of explicit intermediate operations, model-derived procedural guidance becomes more reliable. Appendix D.6 reports a controlled style-pool analysis supporting this task-conditional interpretation.

**Failure modes.** We further analyze failure modes of incorrect solutions by categorizing them into *structural reasoning failures* and *algebraic manipulation errors*. Human guidance predominantly reduces structural failures, while model guidance is more effective at mitigating algebraic errors. SSR reduces both failure types, consistent with executability-aware selection that combines complementary structural and procedural signals (see Appendix D.10 for definitions and quantitative breakdowns).

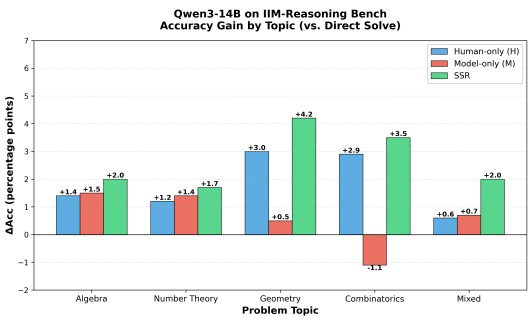

*Figure 6.* Topic-wise gains on HM-REASONINGBENCH using Qwen3-14B. Results exhibit domain-dependent behavior.

### 5.6. Efficiency and Context Budget

We measure output token consumption under DS and SSR using Qwen3-14B across all three benchmarks. Figure 7 shows that SSR reduces total output tokens relative to DS, with reductions concentrated in reasoning tokens. This suggests that SSR lowers *online generation burden* by steering models away from unproductive exploration rather than eliciting longer reasoning traces. Reductions are largest on APEX and HM-REASONINGBENCH, which require longer-horizon reasoning; on AIME25 they are smaller but consistent. This figure should therefore be interpreted as an online token comparison rather than a full end-to-end cost analysis; Appendix D.4 reports the additional offline and online retrieval costs, together with a calibrated comparison to ToT. In particular, SSR trades a modest one-time offline build cost for low online overhead, whereas ToT is training-free but substantially more expensive at test time.

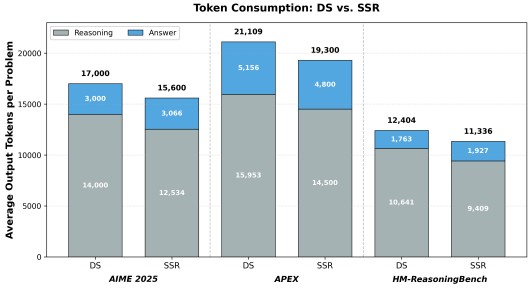

*Figure 7.* Average output token consumption per problem under direct solving (DS) and Selective Strategy Retrieval (SSR) using Qwen3-14B, decomposed into reasoning and final-answer tokens.

**Strategy adherence.** To verify that SSR's gains reflect meaningful strategy execution rather than prompt length, we conduct an adherence-style sanity check. Correct solutions are substantially more likely to correctly instantiate at least one provided strategy, supporting the interpretation that SSR improves executability rather than verbosity. Full protocol

and results are provided in Appendix D.11.

## 6. Conclusion

We revisited example-based guidance for mathematical reasoning from a strategy-level perspective and identified a systematic gap between *strategy usage* and *strategy executability*: strategies that commonly appear in correct solutions are not necessarily those a target model can reliably execute as guidance, explaining the instability of guidance and the limits of uniform imitation.

To address this failure mode, we proposed **Selective Strategy Retrieval (SSR)**, a test-time framework that prioritizes strategies with stronger empirical evidence of executability and consistently outperforms direct solving, in-context learning, and single-source strategy guidance across multiple benchmarks and compact reasoning models.

Additional analyses further sharpen this picture. They show that the usage–executability gap persists on stronger GPT-series models, that executability estimates remain stable under higher-trial evaluation, and that SSR's advantages cannot be reduced to semantic retrieval alone. They also clarify the deployment trade-off of SSR: the method incurs a nonzero but lightweight offline build cost, while keeping online reasoning substantially cheaper than search-based alternatives such as ToT.

Although we study mathematics as a controlled testbed with paired human/model solutions and reliable correctness checking, the underlying distinction between usage and executability is more general. Whenever external guidance must be translated into executable intermediate reasoning by a target model, semantically appropriate strategies may still fail if they are not operationally aligned with that model. Extending this perspective to broader scientific, coding, and logical reasoning domains is a natural next step.

More broadly, our findings suggest that reasoning guidance should be evaluated *model-relatively*: the usefulness of a strategy depends on whether the target model can operationalize it under the given context, motivating executability-aware evaluation and guidance mechanisms grounded in context-dependent effectiveness.

## Impact Statement

This paper presents work whose goal is to advance the understanding of how reasoning guidance interacts with model behavior in mathematical problem solving. We introduce the notion of *strategy executability* to distinguish between reasoning strategies that appear in successful solutions and those that a target model can reliably operationalize when provided as guidance under fixed inference conditions. Building on this perspective, we develop Selective Strategy Retrieval (SSR), a lightweight inference-time framework that improves robustness by selecting and combining strategies based on empirical executability signals rather than surface correctness or prevalence alone.

Beyond its immediate implications for inference-time guidance, this work has potential relevance for future research on model training and evaluation. Our analysis reveals a systematic and structured mismatch between human-written and model-generated solutions: although both may reach correct answers, they exhibit consistent differences in the types of reasoning strategies they employ and successfully execute. This dissociation suggests that current training data distributions and learning objectives may implicitly reinforce certain procedural or algebraic reasoning patterns while under-representing more abstract, structural, or conceptually driven strategies commonly used by humans. From this perspective, human–model disagreement is not merely an inference-time artifact, but a diagnostic signal of imbalance in how reasoning behaviors are learned and reinforced during training.

While this paper does not propose changes to model architectures, training procedures, or supervision schemes, the concept of strategy executability may inform future efforts to design training curricula, auxiliary objectives, or evaluation protocols that better reflect whether models can reliably execute different classes of reasoning strategies under controlled conditions. More broadly, our findings highlight the importance of evaluating reasoning capabilities not only by final-answer correctness, but by the operational usability of intermediate reasoning abstractions.

The expected societal impact of this work is indirect. By clarifying when and why reasoning guidance succeeds or fails, our contributions may support the development of more reliable, interpretable, and controllable reasoning systems for educational, scientific, and analytical applications. This work does not involve human subjects, personal data, or real-world decision-making systems, and it does not introduce new risks beyond those commonly associated with foundational research in machine learning.

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

# A. Implementation Details

## A.1. More Details for HM-ReasoningBench

**Dataset Overview.** HM-ReasoningBench is a large-scale mathematical reasoning benchmark constructed from two complementary sources: *OmniMATH* and *HARP*. After removing exact duplicate problems by text, the final benchmark contains **4,895 unique problems**. Among them, **310 problems** are drawn from HARP, while the remaining **4,585 problems** are sourced from OmniMATH.

**Difficulty Annotation (Level).** Each problem is assigned a discrete difficulty level ranging from Level 1 (easiest) to Level 9 (hardest). In practice, we observe substantial variation in problem difficulty even within the same competition or source, making it insufficient to rely on competition-level tiers or inherited difficulty labels. To obtain a more objective, instance-level assessment, we perform a unified re-annotation of problem difficulty across all sources.

Concretely, difficulty is assigned under a shared reference framework that anchors problems to a common difficulty scale spanning typical Olympiad-style reasoning tasks. GPT-5.1 is used as a calibrated assessor to map individual problems onto this scale, guided by cross-competition comparisons rather than source-specific context. This procedure enforces consistency across heterogeneous sources and enables meaningful cross-source difficulty analysis. As a result, the difficulty distribution concentrates in mid-to-high ranges, with Level 5–7 accounting for the majority of problems.

**Subject Coverage.** Problems are categorized into five broad mathematical subjects: algebra, geometry, number theory, combinatorics, and other. The benchmark is intentionally balanced across core mathematical domains, with combinatorics, number theory, and algebra each accounting for roughly one quarter of the dataset, as shown in table 5.

**Source Characteristics.** The two sources exhibit complementary structural properties. HARP primarily contributes high-difficulty problems, with a strong concentration in Levels 6–7, reflecting its emphasis on advanced multi-step reasoning. In contrast, OmniMATH spans a broader difficulty spectrum from Level 1 to Level 8 and provides wide subject coverage. This combination enables HM-ReasoningBench to support both fine-grained difficulty analysis and robust evaluation of reasoning generalization across problem styles.

**Intended Use.** Overall, HM-ReasoningBench is designed to support fine-grained analysis of mathematical reasoning behaviors across subjects, difficulty regimes, and problem styles. The unified difficulty re-annotation and balanced subject coverage make the benchmark particularly suitable for studying reasoning strategies, cross-domain generalization, and human–model reasoning differences.

## A.2. Strategy Category List

We organize extracted strategies into a fixed set of fine-grained *strategy templates* (i.e., categories), each representing a distinct, recurring reasoning operation (e.g., `angle_chasing`, `modular_arithmetic`, `case_analysis`). For presentation and aggregation, these templates are further grouped into five broad *subjects—Algebraic*, *Number Theory*, *Geometry*, *Combinatorial*, and *Structural*—but all analysis in this paper is conducted at the category level unless stated otherwise.

The complete template list with brief descriptions is provided in Table 6.

*Table 4.* Difficulty level distribution in HM-ReasoningBench after GPT-5.1 re-annotation.

| DIFFICULTY LEVEL | COUNT | PERCENTAGE |
|---|---|---|
| LEVEL 1 | 78 | 1.6% |
| LEVEL 2 | 303 | 6.1% |
| LEVEL 3 | 386 | 7.8% |
| LEVEL 4 | 742 | 15.0% |
| LEVEL 5 | 1140 | 23.1% |
| LEVEL 6 | 1197 | 24.3% |
| LEVEL 7 | 908 | 18.4% |
| LEVEL 8 | 177 | 3.6% |
| LEVEL 9 | 1 | 0.0% |

**Examples of strategy realizations.** To make the template descriptions in Table 6 more concrete, Table 7 lists representative strategy examples.

### A.3. Implementation Details of Selective Strategy Retrieval

This appendix describes the concrete implementation of Selective Strategy Retrieval (SSR), including route-specific candidate selection and ranking. All configurations are fixed across experiments and are not tuned per dataset or model.

**Overview.** SSR retrieves candidate strategies through three routes defined in the main text: Category-Conditioned Retrieval (Route A), Problem-Transfer Retrieval (Route B), and Semantic Fallback Retrieval (Route C). The final candidate pool is formed by taking the union of strategies retrieved from all routes, followed by route-aware ranking.

**Route A: Category-Conditioned Retrieval.** SSR first identifies a small set of compatible strategy categories $\mathcal{C}(x)$ for the target problem. We do *not* train a separate category classifier. Instead, category compatibility is inferred in the same learned graph embedding space used by SSR: each category corresponds to a dedicated node in $\mathcal{G}$, and the graph encoder produces embeddings for problem nodes and category nodes jointly (Appendix A.4).

At test time, given a problem embedding $h_x$, we score each category node $c \in V_c$ by cosine similarity $\mathrm{sim}(h_x, h_c)$ and select the top-2 categories:

$$\mathcal{C}(x) = \mathrm{Top2}_{c \in V_c} \, \mathrm{sim}(h_x, h_c).$$

We then retrieve up to 10 strategies per selected category based on their similarity to $h_x$ within that category, forming a compact set of category-consistent candidates.

**Route B: Problem-Transfer Retrieval.** SSR retrieves strategies that were empirically effective on problems in the neighborhood $\mathcal{N}(x)$. We consider the top 5 most similar problems and collect strategies associated with successful guidance on these problems. This route typically yields a small number of high-precision candidates.

**Route C: Semantic Fallback Retrieval.** When Routes A and B yield insufficient candidates, SSR retrieves additional strategies via semantic similarity. We perform nearest-neighbor search over *strategy node embeddings* produced by the graph encoder (Appendix A.4), using the problem embedding $h_x$ as the query. We retrieve up to 20 strategies. This route is used conservatively and serves only as a fallback.

**Candidate Pool Construction.** Let $\mathcal{S}_A(x)$, $\mathcal{S}_B(x)$, and $\mathcal{S}_C(x)$ denote the strategies retrieved by Routes A, B, and C. The final candidate pool is constructed as

$$\mathcal{S}(x) = \mathcal{S}_A(x) \cup \mathcal{S}_B(x) \cup \mathcal{S}_C(x),$$

with duplicate strategies merged.

### A.4. Executability-Supervised Graph Representation Learning

To support executability-aware strategy selection, we learn structure-aware node representations over the strategy knowledge graph $\mathcal{G}$ using supervised contrastive learning. **This module is not used to estimate executability scores or to directly rank strategies.** Instead, it provides relational features that are later consumed by the supervised executability predictor described in Section 4.4.

**Graph construction.** The heterogeneous graph $\mathcal{G} = (V, E)$ contains three node types: **problems** ($V_p$), **strategies** ($V_s$), and **categories** ($V_c$). Edges encode (i) observed problem–strategy associations extracted from correct solutions in the *training split* of HM-ReasoningBench, and (ii) strategy–category membership. No information from the evaluation split is used in graph construction or supervision, ensuring that all executability signals are strictly confined to training data.

*Table 5.* Subject distribution of HM-ReasoningBench.

| SUBJECT | COUNT | PERCENTAGE |
|---|---|---|
| COMBINATORICS | 1188 | 24.1% |
| NUMBER THEORY | 1167 | 23.7% |
| ALGEBRA | 1164 | 23.6% |
| GEOMETRY | 994 | 20.2% |
| MIXED TOPICS | 419 | 8.5% |

*Table 6.* Strategy taxonomy used throughout the paper. Extracted strategies are mapped to fine-grained templates capturing distinct reasoning operations.

| Subject | Template | Description |
|---|---|---|
| Algebraic | algebraic_general | General symbolic manipulation not covered by specialized algebraic templates. |
| | inequality | Inequality-based reasoning via bounding, convexity, or classical inequalities. |
| | polynomial_analysis | Polynomial structure analysis (factorization, roots–coefficients relations, divisibility). |
| | algebraic_manipulation | Canonical algebraic transformations (substitution, expansion, identity rewriting). |
| | functional_equation | Functional equations and recursive functional constraints. |
| | symmetric_sum | Symmetric polynomial arguments and symmetric-sum identities. |
| Number Theory | modular_arithmetic | Modular reasoning and congruence-based arguments. |
| | prime_factorization | Reasoning via prime decomposition and exponent structure. |
| | divisibility | Divisibility properties and factor-based constraints. |
| | gcd_lcm | GCD/LCM structure and coprimality arguments. |
| Geometry | geometric_general | General geometric reasoning not covered by specialized geometric templates. |
| | angle_chasing | Angle relations derived from geometric theorems and configurations. |
| | circle_properties | Circle geometry (cyclicity, tangency, power of a point, radical axis). |
| | similarity_congruence | Similarity or congruence transformations preserving ratios or lengths. |
| | symmetry_analysis | Exploiting geometric symmetry to simplify structure. |
| | auxiliary_construction | Introducing auxiliary points, lines, or circles to expose hidden relations. |
| | coordinate_general | Coordinate-based or analytic reasoning without an explicit coordinate setup. |
| | coordinate_setup | Explicit coordinate or analytic setup converting geometry into algebraic constraints. |
| | vector_method | Vector-based geometric reasoning (dot/cross products, vector decomposition). |
| | complex_number | Complex-plane representations of geometric transformations. |
| Combinatorial | counting_principle | Direct counting arguments (product/sum rules, recurrences). |
| | inclusion_exclusion | Inclusion–exclusion principle for overlapping sets. |
| | probability_method | Probabilistic reasoning using probability or expectation. |
| | bijection | Establishing bijections to prove counting equivalences. |
| | pigeonhole | Pigeonhole principle and its generalized forms. |
| Structural | extremal_principle | Extremal arguments via minimal or maximal elements. |
| | case_analysis | Structured case partitioning and exhaustive enumeration. |
| | invariant | Invariant or monovariant reasoning under transformations. |
| | proof_by_contradiction | Contradiction-based arguments assuming negation of the claim. |
| | mathematical_induction | Inductive reasoning over integers or recursive structures. |

**Executability supervision.** We obtain supervision from strategy-guided executions on the training split. For each evaluated pair $(x, s)$, we run the target model under a fixed protocol for $T$ independent trials and record outcomes $y_{x,s,1:T} \in \{0, 1\}$. We compute a calibrated executability estimate $\tilde{U}(s \mid x)$ via the Beta–Binomial posterior mean in Eq. (2). Pairs with $\tilde{U}(s \mid x) \geq \delta$ are treated as positives; pairs with $\tilde{U}(s \mid x) \leq \delta^-$ are treated as negatives (we fix $\delta = 0.5$ and $\delta^- = 0.1$ in all experiments), and ambiguous pairs are excluded from contrastive training. Unless otherwise stated, we use $T = 3$ independent decoding trials per $(x, s)$, and sample up to $K = 10$ negatives per positive pair.

**Text encoder for node initialization.** We initialize **problem** nodes and **strategy** nodes with 384-dimensional sentence embeddings from a pretrained SentenceTransformer encoder (we use `all-MiniLM-L6-v2` in all experiments). Category nodes are initialized by mean-pooling the embeddings of strategies assigned to the category. These initial text features are then refined by the graph encoder via message passing.

**Contrastive objective.** For each positive pair $(x, s^+)$, we sample negatives $\mathcal{N}(x)$ from strategies in the same category as $s^+$ that are labeled negative for $x$ (falling back to a global negative pool if necessary). We optimize the InfoNCE loss:

*Table 7.* Representative realizations of strategy templates. Each row provides a neutral action description illustrating how a template may be instantiated in solutions. These examples are for interpretability only and do not affect the taxonomy or experiments.

| Subject | Template | Representative strategy |
|---|---|---|
| Algebraic | algebraic_general | Write down the given quantities and translate all relationships into equations, then solve for the target variable. |
| | algebraic_manipulation | Introduce intermediate variables to simplify expressions or factor a constraint into a product form, then analyze solutions. |
| | polynomial_analysis | Use coefficient extraction (with index shifts) to derive constraints on the coefficient sequence. |
| Combinatorial | counting_principle | Count objects via a direct combination argument (e.g., choose vertices and apply a closed-form expression). |
| | bijection | Analyze injectivity or surjectivity of a mapping and account for collisions by characterizing exceptional patterns. |
| | counting_principle | Correct overcounting caused by symmetry or indistinguishable elements by dividing by the number of equivalent permutations. |
| Geometry | coordinate_setup | Choose a coordinate system and express geometric constraints as algebraic equations, then compute the required quantity. |
| | auxiliary_construction | Add auxiliary points or lines to expose hidden angle relations or similarity structures. |
| | symmetry_analysis | Place the configuration symmetrically (when permitted) to reduce degrees of freedom and simplify the computation. |
| Number Theory | prime_factorization | Restrict candidates via prime factorizations and test feasible exponent patterns systematically. |
| | modular_arithmetic | Establish necessity and sufficiency using congruences, then verify remaining cases directly. |
| | modular_arithmetic | Apply modular constraints to eliminate impossible prime factors before checking exceptional cases. |
| Structural | case_analysis | Partition the problem into cases (e.g., parity or magnitude regimes) and analyze each case exhaustively. |
| | mathematical_induction | Conjecture a closed form from initial cases and prove it by induction using the recurrence relation. |
| | proof_by_contradiction | Assume feasibility and derive a contradiction by comparing magnitudes or signs under the given constraints. |

$$\mathcal{L}_{\text{InfoNCE}} = - \sum_{(x, s^+)} \log \frac{\exp(\text{sim}(h_x, h_{s^+})/\tau)}{\exp(\text{sim}(h_x, h_{s^+})/\tau) + \sum_{s^- \in \mathcal{N}(x)} \exp(\text{sim}(h_x, h_{s^-})/\tau)}. \tag{5}$$

where $\text{sim}(\cdot, \cdot)$ denotes cosine similarity and $\tau$ is a fixed temperature hyperparameter ($\tau = 0.07$). This objective encourages executable problem–strategy pairs to be closer in representation space than non-executable pairs, while controlling for category-level confounds.

**Model architecture.** We use a heterogeneous graph neural network with transformer-based message passing (`TransformerConv`). Separate input projections are applied for each node type (problem, strategy, category), mapping 384-dimensional SentenceTransformer embeddings into a shared hidden space. The network consists of three stacked graph transformer layers with four attention heads, hidden dimension 128, and dropout rate 0.1. Residual connections and layer normalization are applied after each layer.

**Training protocol.** The graph encoder is trained for 50 epochs using the Adam optimizer with learning rate $10^{-3}$ and batch size 32. All hyperparameters are fixed across datasets and target models. The resulting embeddings are used solely as *structural features* for downstream executability prediction, and are not directly used to score or select strategies.

**Sanity check.** To verify that the learned representations capture executability-relevant structure, we evaluate their ability to discriminate executable from non-executable problem–strategy pairs on a held-out subset of the training split. Embedding similarity achieves substantially higher AUC than random baselines, indicating that the contrastive objective encodes meaningful executability information.

# B. Prompt design

## B.1. Strategy Extraction Prompt

This prompt instructs the model to abstract reusable, high-level problem-solving strategies from a given worked solution, focusing on transferable reasoning patterns rather than problem-specific calculations.

---

**Strategy Extraction Prompt**

```
You are a mathematics expert analyzing problem-solving strategies.

Output ONLY valid JSON. Do NOT use markdown or code fences.

Task: extract_solution_strategies

Instructions:
1. From the given solution, extract the KEY STRATEGIES that would help solve
   SIMILAR problems.
2. Write each strategy as a concrete, actionable approach, e.g.:
   - "Express each variable as $p^alpha$ and compare exponents"
   - "Apply inclusion-exclusion to count overlapping cases"

Return the result in the following JSON format:
{
  "strategies": ["strategy1", "strategy2", "..."],
}

Guidelines:
- Limit to 3-5 most critical strategies.
- Focus on reusable reasoning techniques, not problem-specific calculations.
- Be precise, concise, and actionable.

Problem:
{problem_text}

Solution:
{solution_text}
```

---

## B.2. Direct Answer Prompt

This prompt serves as a baseline reasoning setup, asking the model to solve a problem directly without external strategy guidance or example-based hints.

---

**Direct Answer Prompt**

```
Solve the following mathematical problem step by step.

Problem:
{problem_text}

Instructions:
- Provide a detailed solution with clear reasoning.
- Conclude with the final answer.
```

---

## B.3. Strategy Guidance Prompt

This prompt evaluates the effect of explicit strategy-level guidance by providing the model with strategies extracted from similar problems and instructing it to use them during solution construction.

---

**Strategy Guidance Prompt**

```
System: You are an expert mathematician solving competition problems.

User: Solve the following problem using the provided strategy guidance.

Problem:
{problem_text}

Strategy guidance (from similar solved problems):
- {strategy_1}
- {strategy_2}
- ...

Instructions:
- Use the strategies above as hints for your solution approach.
- Solve the problem step by step with clear reasoning.
- Conclude with the final answer.
```

**B.4. Answer Verification Prompt**

This prompt is used to automatically assess the correctness of a model-generated answer by checking mathematical equivalence against a reference solution under strict criteria.

**Answer Verification Prompt**

```
System: You are a rigorous mathematics expert who evaluates student solutions with
    strict standards.

User: You are a rigorous mathematics expert evaluating student answers.

Problem:
{problem_text}

Ground Truth Answer (extracted from reference solution):
{reference_solution}

Student's Final Answer (reasoning process has been separated):
{student_final_answer}

Evaluation task:
Compare the student's final answer with the ground truth answer and determine
whether they are mathematically equivalent.

Mathematical equivalence rules:
- Account for different valid representations:
  * Fractions vs decimals (e.g., 1/2 = 0.5)
  * Mixed numbers vs improper fractions (e.g., 10 2/3 = 32/3)
  * Simplified vs unsimplified forms (e.g., 2/4 = 1/2)
  * Different algebraic forms (e.g., x^2 - 1 = (x-1)(x+1))
  * Equivalent expressions (e.g., 2x + 2 = 2(x+1))
- For proofs: check whether the conclusion is logically equivalent.
- Be STRICT about numerical values (mismatch => wrong).
- Be STRICT about signs (negative vs positive matters).

Classification:
- "Completely Correct": final answer is mathematically equivalent to ground truth
- "Completely Wrong": not equivalent, or missing, or nonsensical

Output ONLY valid JSON (no markdown, no code fences) in this format:
{
  "category": "Completely Correct" | "Completely Wrong",
  "is_correct": true | false,
  "score": 0-100,
  "explanation": "brief explanation"
}
```

## B.5. Strategy Adherence Verification Prompt

This prompt evaluates whether a specific target strategy was actually used—and correctly executed—in a given reasoning trace, enabling fine-grained analysis of strategy executability.

**Strategy Adherence Verification Prompt**

```
You are an expert evaluator of mathematical reasoning strategies.

Task: Determine whether a specific strategy was correctly executed in a student's
    solution.

Problem: {problem_text}

Target Strategy:
Strategy Description: {strategy_text}

Student's Reasoning: {reasoning_excerpt}

Evaluation Instructions:
Assess whether the target strategy was actually used in the student's reasoning,
and if so, whether it was applied correctly.

You must classify the strategy usage into exactly one of the following categories:

1. correctly_executed
    - The strategy was genuinely applied
    - The logical steps follow the methodology of the strategy
    - The execution is correct and contributes to the solution

2. attempted_but_incorrect
    - The student attempted to apply the strategy
    - However, the execution contains logical errors, misapplication, or flawed
        reasoning

3. mentioned_only
    - The strategy is referenced or hinted at via keywords or superficial mentions
    - No substantive application or execution is present

4. not_used
    - There is no evidence that this strategy was used at all

Critical Evaluation Criteria:
- Do NOT rely on keyword matching alone
- Verify that the reasoning steps structurally align with the strategy
- "Correctly executed" requires both correct methodology and correct execution
- Be strict: partial or vague usage should NOT be marked as correctly executed

Output Format:
Return ONLY valid JSON (no markdown, no code fences) in the following format:

{
  "execution_status": "correctly_executed | attempted_but_incorrect |
      mentioned_only | not_used",
  "confidence": 0-100,
  "evidence": "Direct quote from the student's reasoning, or null if not used",
  "explanation": "2-3 sentences explaining the judgment",
  "critical_to_solution": true | false,
  "execution_quality_score": 0-10
}

Important:
Only mark a strategy as "correctly_executed" if there is clear, concrete evidence
that the student followed the intended strategy and applied it correctly.
```

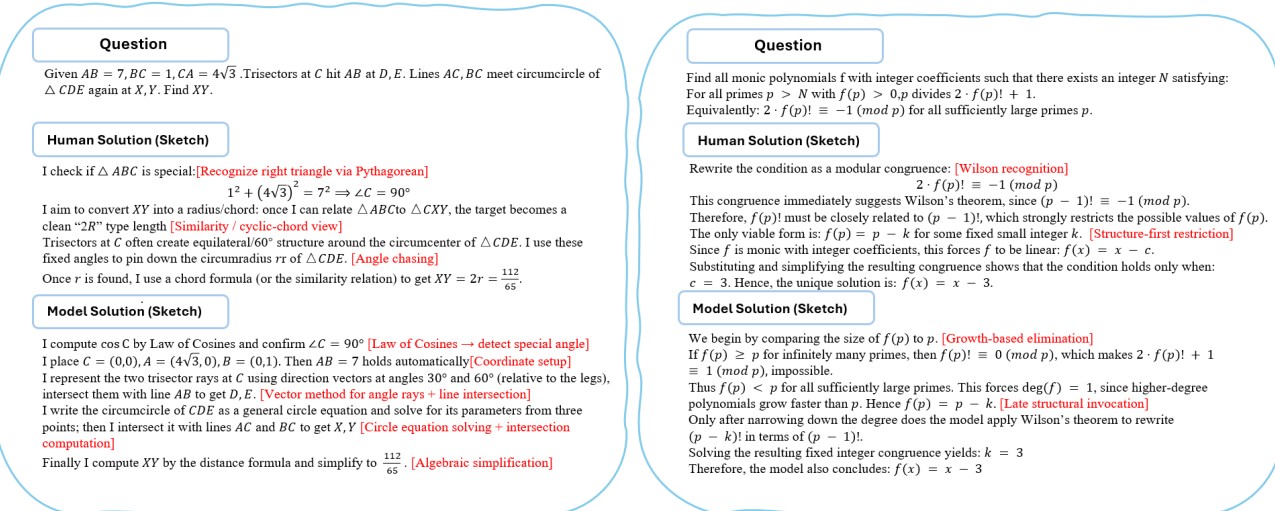

*Figure 8.* Trace-level case studies illustrating strategy executability differences between human-written and model-generated solutions. For each problem, we contrast (i) human-derived strategies, which emphasize structural recognition or theorem-level reasoning, and (ii) model-derived strategies, which rely on procedural or algebraic transformations. Although both solution sources reach the correct final answer, the extracted strategies exhibit different executability properties.

## C. Trace-Level Case Studies of Strategy Executability

Figure 8 provides concrete trace-level illustrations of the strategy divergences analyzed in Section 2. Each example corresponds to a single problem, for which both a human-written solution and a model-generated solution are available and correct.

For each solution, we show the high-level strategies extracted by our pipeline, rather than full step-by-step reasoning. These examples highlight two recurring phenomena observed throughout our analysis. First, human-derived strategies often prioritize early structural recognition (e.g., identifying special geometric configurations or invoking strong theorems), which can be concise but difficult for smaller reasoning models to execute reliably when used as guidance. Second, model-derived strategies tend to emphasize procedural transformations (e.g., coordinate setups or algebraic elimination), which are often more executable but may lack global structure or lead to inefficient reasoning when used alone.

## D. More Experiments

### D.1. Comparison with Stronger Inference-Time Baselines

We compare SSR against several inference-time baselines that improve reasoning by allocating additional test-time computation. All methods use the same base prompt and model backbone as SSR.

For **Self-Consistency (SC)**, we sample $N = 8$ reasoning traces with non-zero temperature and apply majority voting over final answers.

For **Tree-of-Thoughts (ToT)**, we employ a shallow search tree to control inference cost. At each step, the model proposes up to $B = 3$ candidate continuations, and the search is truncated to a maximum depth of $D = 2$. Candidate nodes are scored using a lightweight self-evaluation prompt, where the same model estimates whether a partial reasoning trajectory is likely to lead to a correct solution. At each level, only the top-scoring continuation is expanded further, resulting in at most $1 + B + B$ model calls per problem.

**Least-to-Most Prompting (L2M)** follows the standard decomposition-and-solve procedure described in prior work, where the model first decomposes the original problem into a sequence of simpler subproblems, solves them sequentially, and then composes the final answer.

Table 8 reports the results of these inference-time baselines. Across datasets, SC and L2M provide modest improvements over direct solving on short- and medium-horizon benchmarks. Tree-of-Thoughts yields stronger improvements than SC and L2M under a bounded compute budget, yet remains less stable than SSR, particularly on long-horizon problems. These results indicate that allocating additional test-time computation alone is insufficient; instead, effective guidance depends on selecting strategies that remain executable for the target model.

Notably, SSR achieves these improvements using a single guided generation per problem. In contrast, inference-time baselines such as self-consistency and Tree-of-Thoughts require multiple model calls to sample or search over reasoning trajectories. This suggests that gains from SSR do not arise from increased test-time computation, but from providing strategies that are more executable for the target model.

*Table 8.* Accuracy (%) comparison with inference-time baselines. All methods use the same base prompt and model backbone. Self-Consistency (SC) uses majority voting over $N=8$ samples. Tree-of-Thoughts (ToT) uses branching factor $B=3$ and depth $D=2$.

| | HM-REASONINGBENCH | | | | AIME25 | | | | APEX | | | |
|---|---|---|---|---|---|---|---|---|---|---|---|---|
| MODEL | SC@8 | L2M | ToT | SSR | SC@8 | L2M | ToT | SSR | SC@8 | L2M | ToT | SSR |
| QWEN3-8B | 66.40 | 65.60 | 67.20 | **68.60** | 72.00 | 69.33 | 72.33 | **74.00** | 11.42 | 10.61 | 12.24 | **13.06** |
| QWEN3-14B | 68.80 | 67.40 | 69.60 | **70.20** | 74.33 | 73.33 | **75.00** | 74.67 | 13.88 | 11.83 | **15.91** | 14.69 |
| R1-DISTILL-7B | 50.60 | 48.80 | 51.20 | **52.40** | 48.67 | 45.33 | 51.33 | **53.13** | 8.97 | 7.35 | **9.79** | 7.75 |

## D.2. Retrieval-Based Baselines Beyond ICL

The primary baseline table already includes ICL with a semantically retrieved worked example as a retrieval-based baseline. To more directly compare SSR against retrieval-style alternatives at the *strategy* level, we additionally evaluate several end-to-end retrieval baselines on AIME25; the resulting accuracy comparison is reported in Table 3. Each method uses its own candidate construction and ranking strategy while keeping the same prompt template, hint budget, and decoding protocol as SSR. BM25 retrieves candidates by lexical similarity, Dense retrieval uses embedding-based similarity, and MMR applies diversity-aware semantic reranking before guidance construction.

## D.3. Prior Sensitivity of Executability Estimation

We use a symmetric Beta prior with $\alpha = \beta = 1$ in the main paper. To verify that results are not an artifact of this choice, we vary only the prior in Eq. (2) while keeping the data, target model, decoding protocol, and retrieval pipeline fixed.

*Table 9.* Sensitivity to the Beta prior in Eq. (2) (Qwen3-14B).

| Prior $(\alpha, \beta)$ | Spearman vs. $(1,1)$ | Top-20 overlap | HM-RB | AIME25 | Apex |
|---|---|---|---|---|---|
| $(0.5, 0.5)$ | 0.972 | 93.0% | 70.00 | 74.33 | 14.69 |
| $(1, 1)$ | 1.000 | 100.0% | 70.20 | 74.67 | 14.69 |
| $(2, 2)$ | 0.963 | 91.0% | 69.80 | 74.33 | 14.69 |

The induced rankings remain highly stable, and downstream accuracy varies only slightly, supporting that the usage–executability findings and the SSR gains are not artifacts of a particular weak prior.

## D.4. Full Cost Breakdown and Deployment Trade-Offs

Figure 7 in the main paper compares *online generation* tokens only. Here we add a fuller cost breakdown that includes offline preprocessing/training and online retrieval overhead. We use the same Qwen3-14B setup as in the main paper and compare against ToT with the same bounded-search configuration as Table 8 ($B=3$, $D=2$).

*Table 10.* Average cost per query, averaged across HM-RB, AIME25, and Apex.

| Method | Offline cost (GPU-h) | Online latency (s) | Completion tokens |
|---|---|---|---|
| DS | 0.0 | 7.2 | 16286 |
| ToT ($B=3$, $D=2$) | 0.0 | 27.8 | 35715 |
| SSR | 6.4 | 8.1 | 14114 |

For SSR, the online latency further decomposes into retrieval (0.11s), graph query (0.05s), and guided generation (7.94s). Thus, SSR has nonzero offline cost, but its additional online non-generation overhead is small relative to generation, and its per-query online burden is substantially lower than ToT. Importantly, the offline stage does *not* fine-tune the base reasoning LLM: the trainable SSR modules contain about 0.65M parameters in total (roughly $0.005\%$ of a 14B base model), and there is no gradient update on the base model itself.

## D.5. Feature-Shaping vs. Final Executability Prediction

The graph encoder and the final executability predictor play distinct roles. To verify that contrastive representation learning contributes beyond the logistic predictor alone, we compare a version without contrastive graph representation learning against the full SSR system.

*Table 11.* Ablation of contrastive representation learning (Qwen3-14B, accuracy %).

| Setting | HM-RB | AIME25 | Apex |
|---|---|---|---|
| w/o contrastive representation learning | 68.60 | 70.33 | 13.06 |
| **Full SSR** | **70.20** | **74.67** | **14.69** |

This confirms that contrastive graph learning and supervised executability prediction are distinct but complementary: the graph encoder improves structure-aware features, while the final predictor performs calibrated executability ranking.

## D.6. Task-Conditional Structural vs. Procedural Guidance

To probe the task-conditional interpretation from Section 5.5, we run a controlled style-pool analysis in which retrieval is restricted to either a structural strategy pool or a procedural strategy pool.

*Table 12.* Controlled style-pool comparison (accuracy %).

| Problem family (preferred style comparison) | Qwen3-8B | Qwen3-14B | R1-Distill-7B |
|---|---|---|---|
| Geometry / invariant-driven (Structural vs. Procedural) | 71.67 vs. 67.67 | 74.67 vs. 71.67 | 51.33 vs. 48.67 |
| Counting / modular / algebraic decomposition (Procedural vs. Structural) | 73.33 vs. 68.67 | 75.00 vs. 71.67 | 53.00 vs. 49.67 |

The results support a clear task-conditional pattern: structural guidance helps more when progress depends on the right high-level reformulation, whereas procedural guidance helps more when success depends on explicit intermediate operations.

## D.7. Cross-Model Robustness of the Usage–Executability Gap

To test whether the usage–executability gap is specific to compact open-source models, we repeat the analysis on stronger GPT-series models under the same protocol family. The gap metric is the mean absolute discrepancy between normalized strategy usage and normalized executability over the shared strategy set.

*Table 13.* Capability trend of the usage–executability gap.

| Model | Gap metric | 95% CI |
|---|---|---|
| R1-Distill-7B | 0.238 | [0.214, 0.263] |
| Qwen3-8B | 0.221 | [0.199, 0.243] |
| Qwen3-14B | 0.205 | [0.184, 0.227] |
| GPT-4.1 | 0.186 | [0.166, 0.206] |
| GPT-5.1 | 0.173 | [0.154, 0.193] |

The decreasing trend is significant ($p < 0.01$), but the gap remains clearly non-trivial even for stronger models. High-discrepancy areas remain high-discrepancy as capability increases, including functional equations ($0.31 \rightarrow 0.27 \rightarrow 0.24$), constructive geometry ($0.29 \rightarrow 0.25 \rightarrow 0.22$), and inequality transformation chains ($0.27 \rightarrow 0.23 \rightarrow 0.21$) for Qwen3-8B $\rightarrow$ GPT-4.1 $\rightarrow$ GPT-5.1.

We also analyze whether executability profiles are structurally consistent across model families. Same-series pairs denote Qwen3-8B vs. Qwen3-14B and GPT-4.1 vs. GPT-5.1; heterogeneous pairs are cross-lineage pairs.

*Table 14.* Cross-model consistency of executability profiles.

| Metric | Same-series pairs | Heterogeneous pairs |
|---|---|---|
| Mean Spearman correlation | 0.84 | 0.76 |
| Mean overlap@10 | 0.76 | 0.68 |
| Mean Kendall $\tau$ | 0.66 | 0.57 |

Heterogeneous consistency remains high, indicating a substantial shared cross-model backbone, while same-series consistency is modestly higher, indicating stable lineage-specific shifts. For example, geometric invariant framing is more executable in the GPT series than in the Qwen series ($0.72$ vs. $0.58$), illustrating that the shared backbone coexists with stable model-conditional differences.

### D.8. Robustness of $T = 3$ Executability Estimation

We perform a higher-trial robustness check on a targeted subset of $(x, s)$ pairs, including near-threshold cases, comparing the default $T = 3$ setting against $T = 10$ and $T = 20$.

*Table 15.* Robustness of executability estimation under higher trial counts.

| Comparison | Spearman $\rho$ | Kendall $\tau$ | Flip rate | $\Delta$ downstream EM |
|---|---|---|---|---|
| $T = 3$ vs. $T = 10$ | 0.91 | 0.79 | 5.8% | +0.28 |
| $T = 3$ vs. $T = 20$ | 0.89 | 0.76 | 7.1% | +0.36 |

Rank agreement remains high, binary flips are limited, and moving to much larger trial counts yields only small downstream gains. This supports the use of $T = 3$ as a lightweight but stable executability estimate for the current SSR pipeline.

### D.9. Qualitative Examples

**Example 1: `OmniMATH_2281` (Geometry, Level 2). Problem.** A square is inscribed in a circle of radius 1. Find the perimeter of the square.

**Retrieved Strategies.**

- **(H)** `circle_properties`: Use circle radius/diameter relations (e.g., diagonal equals diameter).

- **(M)** `algebraic_other`: Recall standard formula $P = 4s$.

- ...

**Reasoning Content (simplified).**

A square inscribed in a circle has its diagonal equal to the circle's diameter. Since $r = 1$, the diameter is $2$, so the diagonal is $2$. For a square, $d = s\sqrt{2}$, hence $s = \frac{2}{\sqrt{2}} = \sqrt{2}$. Therefore, the perimeter is $P = 4s = 4\sqrt{2}$. ...

**Content (final answer, simplified).**

Using $d = 2$ and $d = s\sqrt{2}$, we get $s = \sqrt{2}$ and thus $P = 4\sqrt{2}$.

**Example 2: `USAJMO_2013_4` (Number Theory, Level 6). Problem.** Let $f(n)$ be the number of ordered representations of $n$ as a sum of powers of 2. Find the smallest $n > 2013$ for which $f(n)$ is odd.

**Retrieved Strategies.**

- **(H)** `counting_principle`: Characterize when $f(n)$ is odd (often $n = 2^k - 1$).

- **(H)** `mathematical_induction`: Prove the oddness characterization by induction on $k$.

- **(H)** `counting_principle`: Derive recurrence $f(n) = \sum_i f(n - 2^i)$.

- **(M)** `modular_arithmetic`: Reduce to parity by defining $g(n) = f(n) \bmod 2$.

- ...

**Reasoning Content (simplified).**

Define $g(n) = f(n) \bmod 2$ and use the recurrence $g(n) = \sum_{2^k \leq n} g(n - 2^k) \bmod 2$. By computing small cases, we observe $g(n) = 1$ at $n = 1, 3, 7, 15, \ldots$, suggesting $f(n)$ odd $\iff n = 2^k - 1$. The next such number after 2013 is $2^{11} - 1 = 2047$. ...

**Content (final answer, simplified).**

Since $f(n)$ is odd $\iff n = 2^k - 1$, the smallest $n > 2013$ is $2^{11} - 1 = 2047$.

**Example 3: `OmniMATH_3827` (Algebra, Level 5). Problem.** Find all functions $f : \mathbb{R} \to \mathbb{R}$ such that $f(xy) = f(x)f(y) + f(f(x + y))$ for all $x, y \in \mathbb{R}$.

**Retrieved Strategies.**

- **(H)** `functional_equation`: Verify candidate functions by direct substitution.

- **(M)** `case_analysis`: Plug in special values ($x = 0, 1$, $y = 0$, etc.) to derive constraints.

- ...

**Reasoning Content (simplified).**

Set $x = 0$ to obtain $f(0) = f(0)f(y) + f(f(y))$, hence $f(f(y)) = c(1 - f(y))$ where $c = f(0)$. Test constant solutions: $f \equiv 0$ works. Assume affine form $f(x) = ax + b$ and compare coefficients, yielding $a \in \{0, 1\}$, and the nonzero affine solution $f(x) = x - 1$. Finally, verify by substitution that $f(x) = 0$ and $f(x) = x - 1$ satisfy the equation. ...

**Content (final answer, simplified).**

The only solutions are $f(x) \equiv 0$ and $f(x) = x - 1$.

### D.10. Failure-Mode Analysis

We decompose incorrect solutions into two broad failure modes: (i) *algebraic manipulation errors*, where the global solution plan is largely correct but execution fails due to symbolic or arithmetic mistakes; and (ii) *structural reasoning failures*, where the solution fails to establish or exploit the correct global structure, such as missing a key decomposition, invariant, or case split.

Figure 9 reports the distribution of failure modes on HM-REASONINGBENCH for Qwen3-14B. Human guidance primarily reduces structural failures, reflecting its strength in providing global organization and conceptual structure. In contrast, model guidance more effectively reduces algebraic manipulation errors. SSR mitigates both failure modes, consistent with executability-aware selection that combines complementary structural and procedural signals.

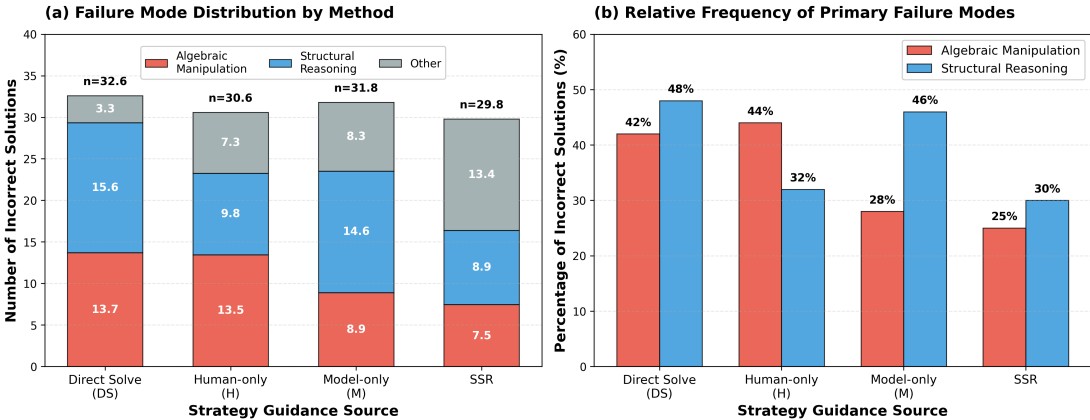

*Figure 9.* Failure-mode analysis on HM-REASONINGBENCH using Qwen3-14B. Human guidance primarily reduces structural failures, model guidance reduces algebraic errors, while SSR mitigates both error types.

### D.11. Strategy Adherence Evaluation Protocol

This appendix provides implementation details for the strategy adherence sanity check reported in Section 5.6, and summarizes the corresponding results in Figure 10.

**Setup.** We randomly sample 100 problems from the HM-ReasoningBench test split and evaluate three models: DeepSeek-R1-Distill-Qwen-7B, Qwen3-8B, and Qwen3-14B. For each problem, SSR provides up to five abstract strategy hints as guidance. Each model generates a single solution under the same prompting and decoding configuration used in the main experiments.

**Adherence criterion.** We do not require the model to explicitly mention a strategy or follow it verbatim. A strategy is considered *correctly executed* if the generated solution applies the strategy in a way that substantively contributes to a valid solution. Superficial mentions or partial but incorrect applications are not counted as execution.

**Evaluation procedure.** We use GPT-5.1 as an independent evaluator. The evaluator is provided with (i) the model-generated solution and (ii) the list of strategy hints given as guidance, and outputs a binary judgment for each strategy indicating whether it is correctly instantiated in the solution. The evaluator is instructed to assess functional correctness rather than textual overlap. The full evaluation prompt is provided in Appendix B.5.

**Metrics.** For each solution, we compute: (i) the number of correctly executed strategies, and (ii) a binary indicator of whether at least one strategy is correctly executed. Statistics are aggregated separately over correct and incorrect final answers.

**Results summary.** Figure 10 reports adherence statistics as a function of model size, separately for correct and incorrect final answers. Across models, correct solutions exhibit both a higher number of correctly executed strategies and a substantially higher probability of executing at least one strategy, with the gap widening for larger models.

**Interpretation.** Incorrect solutions often exhibit exploratory reasoning that may superficially touch on multiple strategies without successfully applying any of them. The proposed metrics therefore focus on whether the provided guidance enables at least one strategy to be operationalized correctly, directly supporting the executability-based interpretation discussed in Section 5.6.

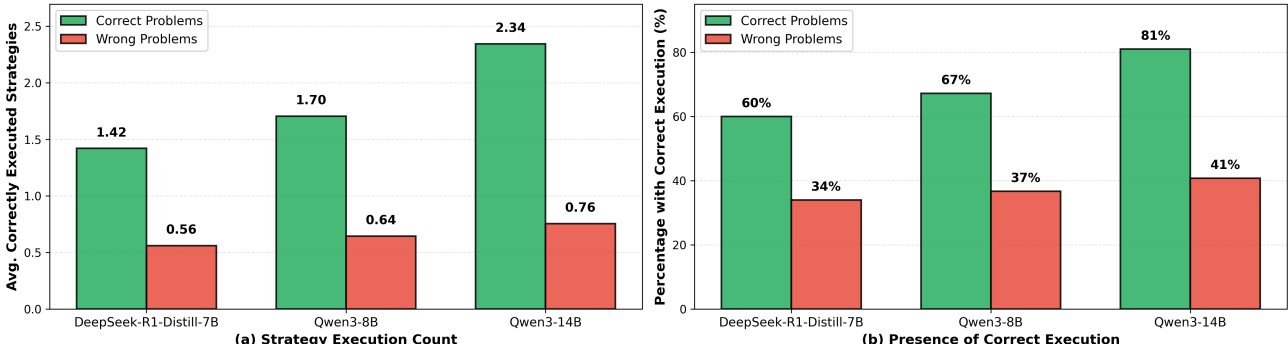

*Figure 10.* Strategy adherence analysis. **Left:** average number of strategies correctly executed in the generated solution. **Right:** percentage of problems for which at least one provided strategy is correctly executed. Results are reported separately for correct and incorrect final answers. Correct solutions consistently exhibit stronger strategy execution, with the gap widening for larger models.

