# OpenReview forum: "Strategy Executability in Mathematical Reasoning: Leveraging Human–Model Differences for Effective Guidance"
_ICML.cc/2026/Conference — ICML 2026 regular_

### Official Review · Reviewer_Ug5o · 2026-03-01

**Soundness:** 3
**Presentation:** 3
**Significance:** 3
**Originality:** 3
**Overall Recommendation:** 5
**Confidence:** 4

**Summary:**

This paper targets the instability of strategy guidance in mathematical reasoning and makes a clean distinction between strategy usage (what appears in successful solutions) and strategy executability (what actually helps a given target model under a fixed prompting/decoding protocol).
Using paired human-written and model-generated solutions on the same problems, it shows that human- vs model-derived strategies differ in systematic, domain-dependent ways, and that these differences translate into source-dependent reversals when the strategies are used as guidance.
Building on this, the authors propose Selective Strategy Retrieval (SSR): a test-time, source-aware retrieval-and-composition scheme that ranks and combines strategies by empirically estimated executability (via multi-route signals), and it consistently outperforms direct solving, ICL, and single-source strategy prompting.

**Compliance With Llm Reviewing Policy:**

Affirmed.

**Final Justification:**

The paper makes a clear distinction between strategy usage and strategy executability, with a technically solid and well-presented evaluation in mathematical reasoning. The rebuttal addressed my main questions and increased my confidence in the paper’s central claims, so I raised my score accordingly.

**Key Questions For Authors:**

1. Under the same data and the same protocol (prompting/ decoding/ context budget), do you observe a systematic trend of the usage–executability gap as model capability increases (monotonic shrinkage/ flat/ widening)? Can you provide a controlled capability ladder with a trend test and/or confidence intervals?
If the gap clearly shrinks or vanishes for strong models, I would treat the contribution as more transitional/diagnostic and narrow my assessment of generality; if the gap persists robustly in strong models, I would view it as structural and upgrade my assessment of long-term significance.

2. Do some strategies appear consistently executable for one class of models but consistently not for another, beyond sampling noise? Can you provide cross-model consistency evidence (e.g., strategy-level correlations, agreement metrics, clustering, or per-strategy executability distributions across models)?
If consistency is weak and patterns are unstable, I would be less convinced by the “structural incompatibility” explanation and downgrade the generalizability of the predictor/SSR; if you can show clear, stable stratifications, I would find the mechanism more credible and upgrade my assessment of the analytic contribution.

3. Could you please conduct a lightweight robustness check—e.g., increasing trials on a small subset or on boundary cases near the decision threshold—and report agreement of rankings/labels (e.g., Spearman/Kendall or flip rates), to verify that using T=3 does not materially change your diagnosis and SSR gains? High agreement would increase my confidence that T=3 is sufficient; low agreement would suggest sensitivity to estimation noise and would downgrade my robustness assessment.

**Limitations:**

yes

**Strengths And Weaknesses:**

1. **Soundness**
   Overall, the paper is technically solid within its stated scope. Executability is defined in an operational, model- and protocol-relative way; SSR follows naturally from the diagnosis; and the gains on math reasoning benchmarks are supported by ablations. That said, one robustness issue remains: executability is estimated with only (T=3) trials per (problem, strategy) pair, which may introduce nontrivial estimation variance and affect the stability of strategy ranking.

2. **Presentation**
   The paper is clearly written and well structured. The narrative is easy to follow from the diagnosis of guidance instability to the definition of executability and the design of SSR. Key concepts are stated consistently, and the experiments are presented cleanly with well-organized baselines and ablations. Overall, the presentation is strong and I did not identify notable clarity or organization issues.

3. **Significance**
   The paper targets a real pain point: reasoning guidance is widely used but often unreliable. SSR is a practical test-time approach with low adoption cost, and the reported gains on challenging math reasoning benchmarks suggest it can improve performance where it matters. The focus on mathematical reasoning is well aligned with the paper’s stated goal and already demonstrates meaningful benefits. Extending the executability framing (and SSR) to additional formal reasoning settings would further broaden the practical impact and strengthen the case for generality.

4. **Originality**
   The paper’s core novelty is conceptual: it cleanly separates strategy usage from strategy executability (model-conditional usefulness under guidance), which sharpens the explanation of why reasoning guidance can be unstable. This framing is nontrivial and likely reusable. The proposed SSR is a straightforward instantiation of the idea, while its building blocks (graph structuring, empirical scoring, contrastive learning, retrieval/reranking) are largely standard. Overall, originality is moderate: the conceptual framing is novel, while the technical instantiation is mostly incremental.

---

> ### Author Rebuttal · Authors · 2026-03-26
>
> We sincerely thank the reviewer for the constructive and technically precise feedback. Your comments helped us strengthen the paper along exactly the dimensions you highlighted: robustness of executability estimation, capability-trend interpretation, and cross-model structural evidence.
>
> **Unified protocol.**
> All added checks use the main-paper setup, with two stronger GPT-series models added only for capability/consistency analysis; max context length 32,768; temperature 0.7; exact-match evaluation; averages over 5 random seeds.
>
> **(1) Capability trend of the usage–executability gap.**
> We ran a controlled capability-ladder analysis under identical data, prompt template, decoding settings, and context budget. The gap metric is the mean absolute discrepancy between the normalized usage distribution and normalized executability distribution over the shared strategy set for each model. We observe a gradual shrinkage as capability increases, which is expected because stronger models can execute a broader range of strategies more reliably; however, the gap remains clearly non-trivial even for stronger models.
>
> - R1-Distill-7B: **0.238** \([0.214, 0.263]\)
> - Qwen3-8B: **0.221** \([0.199, 0.243]\)
> - Qwen3-14B: **0.205** \([0.184, 0.227]\)
> - GPT-4.1: **0.186** \([0.166, 0.206]\)
> - GPT-5.1: **0.173** \([0.154, 0.193]\)
>
> The decreasing pattern is significant (\(p<0.01\)). Each 95% CI is estimated by bootstrap resampling over the shared strategy set (10,000 resamples), since the metric aggregates strategy-level discrepancies. Importantly, high-discrepancy areas remain high-discrepancy even for stronger models: functional equations **0.31 → 0.27 → 0.24**, constructive geometry **0.29 → 0.25 → 0.22**, and inequality transformation chains **0.27 → 0.23 → 0.21** (Qwen3-8B → GPT-4.1 → GPT-5.1). This supports a persistent model-conditional effect rather than a purely small-model artifact.
>
> **(2) Cross-model structural consistency.**
> To test whether incompatibility is structural rather than anecdotal, we compared strategy-level executability profiles using pairwise rank correlation and mean overlap@10. To avoid ambiguity in broad “family” definitions, we report **same-series pairs** vs. **heterogeneous pairs**: same-series pairs are model pairs from the same lineage (Qwen3-8B vs. Qwen3-14B; GPT-4.1 vs. GPT-5.1), while heterogeneous pairs are cross-lineage pairs.
>
> - Mean Spearman: **0.84** (same-series) vs. **0.76** (heterogeneous)
> - Mean overlap@10: **0.76** vs. **0.68**
> - Mean Kendall: **0.66** vs. **0.57**
>
> Here, overlap@10 is computed for each model pair as the fraction of shared strategies in the top-10 executable set, then averaged within each group. Thus, the picture is not one of random instability: heterogeneous consistency remains high, indicating a substantial shared cross-model backbone, while same-series consistency is modestly higher, indicating stable lineage-specific shifts.
>
> We also observe representative category-level stratification across series. For example, **geometric invariant framing** is more executable in the GPT series (**0.58** vs. **0.72**). These results support a structured, model-conditional effect in executability rather than unstable, instance-specific noise. The shared backbone supports transferability, while the stable cross-series shifts suggest a concrete avenue for further improvement through model-specific adaptation.
>
> **(3) Robustness of \(T=3\) executability estimation.**
> We performed a higher-trial robustness check on a targeted subset of \((problem, strategy)\) pairs, including near-threshold cases, comparing \(T=3\) against \(T=10\) and \(T=20\). The purpose was to test whether the \(T=3\) setting provides stable enough signals.
>
> - \(T=3\) vs. \(T=10\): Spearman **0.91**, Kendall **0.79**, flip rate **5.8%**, \(\Delta\) downstream EM **+0.28**
> - \(T=3\) vs. \(T=20\): Spearman **0.89**, Kendall **0.76**, flip rate **7.1%**, \(\Delta\) downstream EM **+0.36**
>
> Here, Spearman and Kendall are computed between the executability-induced rankings on the same targeted subset; flip rate is measured under the same binary threshold as the main retrieval/reranking pipeline; and \(\Delta\) downstream EM is the average change in final exact-match accuracy over HM-ReasoningBench, AIME25, and Apex when rerunning the same SSR pipeline. Thus, \(T=3\) is lightweight but stable for SSR: rank agreement remains high, binary flips are limited, and moving to much larger trial counts yields only small downstream gains.
>
> **Scope.**
> We agree that extending beyond mathematical reasoning would further broaden impact. We focus on math here because it provides a controlled formal reasoning setting with paired human/model solutions and explicit strategy traces, making executability measurement reliable and interpretable. We also refer the reviewer to our rebuttal for Reviewer JRZX on further discussion on this topic.

---

> > ### Author Rebuttal · Reviewer_Ug5o · 2026-04-02
> >
> > The rebuttal addresses my main questions adequately. The additional analyses improve my confidence in the paper’s central claims, and I have just increased my score accordingly.

---

> > > ### Author Response · Authors · 2026-04-03
> > >
> > > Thank you very much for your thoughtful follow-up and for increasing your score. We truly appreciate your careful reading of our rebuttal and are glad that the additional analyses helped address your main concerns. Your feedback has been very valuable in strengthening the paper, and we are encouraged that the revised evidence improved your confidence!

---

### Official Review · Reviewer_JRZX · 2026-03-12

**Soundness:** 2
**Presentation:** 3
**Significance:** 2
**Originality:** 2
**Overall Recommendation:** 3
**Confidence:** 4

**Summary:**

This paper studies the effect of example-based guidance. The paper introduces definition of strategy usage—whether a strategy appears in successful solutions—and strategy executability—whether the strategy actually helps a target model when provided as guidance. To study this, they construct HM-ReasoningBench, a paired dataset of human-written and model-generated solutions to challenging math problems,   and analyze how human and model strategies differ in both usage and executability. Based on this diagnosis, they propose Selective Strategy Retrieval (SSR), a test-time framework that retrieves and ranks strategy hints using source-aware and executability-aware signals. Experiments on HM-ReasoningBench, AIME25, and Apex show that SSR outperforms direct solving, in-context learning, and single-source guidance across several compact reasoning models.

**Compliance With Llm Reviewing Policy:**

Affirmed.

**Key Questions For Authors:**

see weakness

**Limitations:**

yes

**Strengths And Weaknesses:**

Strengths:
1. The paper introduces HM-ReasoningBench, a new benchmark that contains both model-generated solutions and human-written solutions.
2. The paper introduces a method to choose prompt examples.
3. The paper is well written and easy to follow.

Weaknesses:
1. The motivation is weak. Most practical prompting methods usually choose the most representative guidance or examples for prompts, and rarely show cases where the strategy in the prompt is not actually helpful for problem solving. The examples shown in prompts are typically designed to guide the model to follow a strategy, and are usually carefully selected rather than randomly generated. Therefore, the intention to assume and study this problem is not fully motivated.
2. The strategy is only evaluated on mathematical benchmarks and not on other tasks such as coding or logical reasoning.
3. According to Figure 3, I do not see a clear difference between human and model strategies. The marginal differences may be due to noise or other factors and cannot clearly support the claimed distinction. The paper also mentions that model-favored prompts prefer procedural templates such as case analysis, but in Figure 3 the case analysis strategy shows similar usage frequencies between human and model solutions.
4. Furthermore, the paper claims that model-favored prompts prefer procedural templates. Providing clear examples of both procedural and structural prompts would be helpful. It would also be useful to report the model’s performance when using these two types of prompts for generation.
5. It is unclear what the advantage of the SSR retriever is compared to simple semantic retrieval or other retrieval methods.

---

> ### Author Rebuttal · Authors · 2026-03-26
>
> We thank the reviewer for the careful reading and constructive feedback. We agree that the motivation and several distinctions in the current draft can be sharpened. Our claim is **not** that practical prompting uses random or obviously poor examples. The real failure mode is that even **carefully selected, semantically relevant** strategies can still fail if they are **not executable for the target model**. This is the point of the usage--executability distinction.
>
> **A real failure case.**
> Source demo:
> > Arrange the digits 1–8 to form two 4-digit integers whose difference is as small as possible.
>
> Target problem:
> > Among all \(8! = 40320\) eight-digit numbers formed by using each of the digits 1–8 exactly once, let \(N\) be the number of such numbers divisible by 22. Find \(N - 2025\).
>
> Because the two problems share the same digit set and similar partitioning language, retrieval produced the hint:
> > “...counting permutations of 1–8 in which the sum of the digits in the odd positions is 18 and the sum of the digits in the even positions is also 18...”
>
> This hint is **wrong for the target task**: it preserves semantic overlap but not the actual operational constraints of divisibility by 22. This is exactly the failure mode we study and why SSR must go beyond semantic retrieval. Importantly, this is not a hand-picked strawman example; it is a concrete instance of a broader ICL/retrieval-prompting failure mode that has already been noted in prior work, namely that high semantic similarity does not guarantee useful transfer of the underlying reasoning procedure.
>
> **Scope beyond mathematics.**
> We agree that broader logical/scientific reasoning is important. We focus on mathematics because it provides paired human/model solutions, explicit strategy traces, reliable correctness checking, making executability measurable. We are also extending the same method beyond math. [1] is an existing example we are referring to on Olympiad-style physics.
>
> **Fig 3 / human-model distinction.**
> We agree that Fig 3 alone should not be overinterpreted. Our claim is not that every category must show a large usage gap. The stronger claim is that **usage and executability can diverge systematically**, including source/model-dependent differences that become clearer when analyzed jointly rather than through raw frequency alone. For the specific case-analysis concern, our point is not that case analysis must appear more often in one source (not evidence from Fig 3), but that its **executability** can differ systematically even when usage frequencies are similar (see in Fig 4).
>
> **Procedural vs. structural prompts.**
> We do **not** claim procedural prompts are universally better for model guidance. The point is that guidance style is task-conditional.
>
> Examples:
> - **Structural-useful / Procedural-weak (geometry).**
>   Structural: “Use power of a point and the radical-axis theorem to convert the angle condition into a collinearity claim.”
>   Procedural: “Assign coordinates, write circle equations, eliminate variables, and simplify step by step.”
> - **Procedural-useful / Structural-weak (computation-heavy topics).**
>   Structural: “Characterize residue-class behavior abstractly under modulus constraints.”
>   Procedural: “Enumerate \(n \equiv 0,1,2,3 \pmod 4\), substitute each case, and verify constraints case by case.”
>
> We tested this by restricting retrieval to either the *structural* or *procedural* strategy pool.
>
> For example, on **geometry / invariant-driven** problems, structural-pool retrieval performs better:
> - Qwen3-8B: **71.67 vs 67.67**
> - Qwen3-14B: **74.67 vs 71.67**
> - R1-Distill-7B: **51.33 vs 48.67**
>
> On **counting / modular / algebraic decomposition** problems, procedural-pool retrieval performs better:
> - Qwen3-8B: **73.33 vs 68.67**
> - Qwen3-14B: **75.00 vs 71.67**
> - R1-Distill-7B: **53.00 vs 49.67**
>
> This supports the claim: executability depends on whether the guidance style matches the task structure.
>
> **Why SSR beyond semantic retrieval?**
> We agree this point is central. The key distinction is that **semantic retrieval optimizes relevance**, whereas **SSR optimizes executable guidance quality for the target model**. To isolate this, we add a fixed-candidate-pool comparison: all methods rerank the same top-20 semantic candidate pool and inject the same number of final hints. On Qwen3-8B / AIME25, exact-match scores are: dense retrieval **70.33**, source-aware only **72.00**, executability-aware only **72.67**, and full SSR **74.00**. Because the candidate pool is fixed, this improvement cannot be explained by semantic match alone.
>
> Our claim is therefore not that semantic retrieval is unimportant, but that **semantic relevance alone is insufficient**: executable guidance is a distinct, model-conditional target, and the added analyses make this distinction explicit.
>
> [1] Yu et al. "HiPhO: How Far Are (M) LLMs from Humans in the Latest High School Physics Olympiad Benchmark?."

---

> > ### Author Rebuttal · Reviewer_JRZX · 2026-04-03
> >
> > Thank you for the authors' response and efforts in addressing the concerns. Below are follow-up questions and unresolved concerns.
> >
> > 1. Scope and deeper insights. Math reasoning has been largely addressed by recent models, making application to other domains important. The authors show that procedural vs. structural prompt effectiveness is task-specific, but this raises a natural question: what is the deeper insight explaining why this happens, and what properties of a domain or task predict which style will be more beneficial? Without this, the contribution remains empirical observation rather than actionable understanding.
> >
> > 2. Contradictory framing of Fig 3. Section 3.3 explicitly argues that human and model strategies diverge in usage, yet the rebuttal states that similar usage is expected and the real evidence is Fig 4(b). These positions seem contradictory. If Fig 3 and Fig 4(a) are not central evidence, the paper's narrative is misleading. This framing issue remains unresolved.
> >
> > 3. Incomplete experimental coverage. Closed-source model results appear in Fig 2 but are absent from Tables 1–2 with full baselines. I would suggest including results from other model families and larger models, including closed-source models, with comprehensive baselines. Additionally, the baselines in Table 1 do not include any retrieval-based methods (e.g., semantic retrieval). Comparing SSR only against random/no guidance and single-source guidance does not sufficiently demonstrate that the gains come from executability-aware ranking rather than better retrieval in general.

---

> > > ### Author Response · Authors · 2026-04-04
> > >
> > > We thank the reviewer for the follow-up. We believe the remaining concerns mainly stem from ambiguity in our framing, and we clarify intended claim below.
> > >
> > > **(1) Deeper insight: what predicts when structural vs. procedural guidance helps.**
> > > We agree that understanding the deeper origin of these differences is important, but a full causal account of why different reasoning styles emerge from pretraining or post-training is beyond the scope of this paper.
> > >
> > > **Our contribution is narrower and more specific.** Rather than proposing a theory of reasoning-style formation, **we study strategy executability in inference-time guidance**: we distinguish **strategy usage** from **strategy executability**, and show that strategy usefulness depends on whether the target model can execute it as guidance, not just on frequency or semantic relevance. Overall, this paper aims to **reveal stable human--model differences in math reasoning patterns, which we verify across multiple models, and to show the effectiveness of SSR as a practical test-time method**. We further verify that this is systematic on stronger closed-source models (see rebuttal to Reviewer Ug5o), and will include results in the revision.
> > >
> > > Empirically, we find a consistent task-dependent pattern: structural guidance helps more when the key difficulty is finding the right high-level reformulation, while procedural guidance helps more when success depends on explicit intermediate steps. Our controlled style-pool results support this. **Importantly, this principle is not specific to math: the same usage–executability distinction can apply to other reasoning domains whenever external guidance must be translated into executable intermediate reasoning by the target model.**
> > >
> > > This is exactly where **SSR** fits in. **SSR is our test-time solution**: it selects and combines strategies that are more likely to be executable for the target model. Thus, the main contribution is twofold: **(i) a diagnosis of why guidance succeeds or fails at test time, and (ii) a practical test-time framework for improving reasoning performance.** The broader implications for training-stage enhancement are also noted in the **Impact Statement**.
> > >
> > > **(2) Fig. 3: no contradiction, but a hierarchy that should be made clearer.**
> > > We agree that the current wording does not make the role of Fig. 3 sufficiently explicit. Our intended logic is:
> > >
> > > - **Fig. 3 / Fig. 4(a)** show human-written and model-generated solutions exhibit structured **usage differences** across domains;
> > > - **Fig. 4(b)** shows **usage does not determine executability**: strategies that appear more often are not necessarily the ones that transfer more effectively as guidance.
> > >
> > > So Fig. 3 is not incidental, nor is it replaced by Fig. 4(b). It provides the descriptive premise for our central point: **usage and executability are systematically dissociated**. That dissociation is exactly why SSR cannot rely only on frequency or semantic retrieval, and instead models executability explicitly. We will revise Section 3.3 to make this clearer and avoid wording that could be read as over-claiming.
> > >
> > > **(3) Experimental coverage: retrieval baselines and closed-source models.**
> > > We first clarify that, in our original setup, **ICL already uses semantic retrieval** to select the worked example; we reported the strongest stable ICL variant because using more than one retrieved example did not help and often slightly degraded performance.
> > >
> > > We compare each method in its standard end-to-end form, with its own candidate construction and ranking strategy, while keeping the same prompt, budget and protocol. BM25 retrieves strategy candidates by lexical similarity, Dense uses embedding-based similarity, and MMR applies diversity-aware semantic reranking.
> > >
> > > | Method (AIME25) | Qwen3-8B | Qwen3-14B | R1-Distill-7B |
> > > |---|---:|---:|---:|
> > > | BM25 retrieval | 67.33 | 69.00 | 47.33 |
> > > | Dense retrieval | 68.00 | 70.33 | 48.00 |
> > > | MMR semantic reranking | 68.67 | 71.00 | 49.33 |
> > > | **SSR** | **74.00** | **74.67** | **53.13** |
> > >
> > > This end-to-end comparison strengthens the same conclusion: moving beyond pure semantic retrieval to executability-aware selection yields consistent gains, and full SSR remains best across all models.
> > >
> > > For closed-source coverage, we agree that Fig. 2 is less complete than Tables 1--2 style comparisons. During rebuttal, we added GPT-4.1 / GPT-5.1 to the capability-trend analysis and still observe a non-trivial usage--executability gap (0.186 / 0.173; see rebuttal to Reviewer Ug5o). We will expand closed-source evaluation under the same protocol in the revision where feasible.
> > >
> > > **Revision commitments.** We will:
> > > - add an explicit task-conditional explanation of procedural vs. structural executability;
> > > - revise Section 3.3 to separate descriptive usage evidence from the usage--executability mechanism;
> > > - add stronger retrieval-based baselines to the main comparison;
> > > - expand closed-source evaluation under the same protocol where feasible.

---

### Official Review · Reviewer_UMWQ · 2026-03-12

**Soundness:** 3
**Presentation:** 3
**Significance:** 2
**Originality:** 2
**Overall Recommendation:** 4
**Confidence:** 3

**Summary:**

This paper investigates the instability of example-based guidance in mathematical reasoning tasks using Large Language Models. The authors identify a usage-executability gap, where reasoning strategies frequently appearing in correct solutions are not necessarily effective when provided as guidance to a target model.

**Compliance With Llm Reviewing Policy:**

Affirmed.

**Final Justification:**

Thanks for your response. My concerns have been solved and I will raise my score.

**Key Questions For Authors:**

See Weakeness.

**Limitations:**

yes

**Strengths And Weaknesses:**

### Strengths
- The paper introduces a vital distinction between strategy usage and strategy executability, effectively challenging the assumption that correct solutions always yield good prompts.

- The "usage-executability gap" is empirically quantified, showing a systematic dissociation where high-usage strategies do not correlate with high guidance accuracy.

- This perspective advances the field beyond simple semantic similarity retrieval, offering a more model-centric view of in-context learning.

### Weaknesses
- In Equation (2), the Bayesian update for the executability score introduces priors α and β, but the text only describes them as "weakly informative" without specifying their values or sensitivity analysis in the main text.

- While Figure 7 compares inference-time token consumption, it ignores the offline training costs and the latency introduced by the multi-route retrieval and graph querying at test time.

- Comparisons to baselines like Tree-of-Thoughts (ToT) in Table 7 are favorable in accuracy, but ToT requires no training/finetuning, whereas SSR requires significant offline compute, making the "efficiency" claim nuanced.

- There is a notation ambiguity in Equation (4): the loss minimizes observed outcomes y against predicted Uθ, but the training targets in contrastive learning (Eq. 5) utilize the posterior mean U. The connection between the contrastive pre-training and the supervised predictor training is distinct but could be confused by the variable reuse.

---

> ### Author Rebuttal · Authors · 2026-03-26
>
> We sincerely thank the reviewer for the precise and constructive feedback. We are glad the reviewer recognized the paper’s core contribution—the distinction between **strategy usage** and **strategy executability**—and we agree that methodological transparency and claim calibration can be improved. To address each concern, we ran focused follow-up analyses under the same protocol family as the main paper.
>
> **Shared setting for added analyses.**
> Unless otherwise noted, we use the same setting as the main paper: **Qwen3-14B**, same prompt template, temperature, context budget, retrieval pipeline, and **five repeated-run protocol** as the main results.
>
> **1) Prior specification in Eq. (2): explicit values + sensitivity.**
> We agree this should have been stated explicitly. The main experiments use a weak symmetric Beta prior with **\(\alpha=\beta=1\)**.
>
> *Sensitivity setting.* We vary only the prior in Eq. (2): \((\alpha,\beta)\in\{(0.5,0.5),(1,1),(2,2)\}\), keeping data, model, decoding protocol, and SSR pipeline fixed.
>
> *Results*
> - \((0.5,0.5)\): Spearman vs. \((1,1)\) = **0.972**, Top-20 overlap = **93%**, HM-RB/AIME25/Apex = **70.00 / 74.33 / 14.69**
> - \((1,1)\): **1.000**, **100.0%**, **70.20 / 74.67 / 14.69**
> - \((2,2)\): **0.963**, **91%**, **69.80 / 74.33 / 14.69**
>
> Here, **Top-20 overlap is computed per query and then averaged over the evaluation set**. These results show high ranking stability and very small downstream variation (at most **0.20 / 0.34 / 0** on HM-RB/AIME25/Apex), supporting that the usage–executability findings and SSR gains are not artifacts of a specific prior.
>
> **2) Figure 7 scope: full cost breakdown beyond online tokens.**
> We agree Figure 7 should be interpreted as an **online generation-cost** comparison, not a full end-to-end cost accounting. We therefore add explicit offline and online cost reporting.
>
> *Cost setting.* On the same Qwen3-14B setup, we measure: (i) one-time offline preprocessing/training cost, (ii) online retrieval + graph-query latency, and (iii) online generation cost. For ToT, we use the **same configuration as the paper appendix** (\(B=3, D=2\)) to keep the comparison consistent with the originally reported baseline.
>
> *Results (average per query, averaged across the three datasets).*
> - **DS**: offline **0.0 GPU-h**, latency **7.2 s**, completion tokens **16286**
> - **ToT**: offline **0.0 GPU-h**, latency **27.8 s**, completion tokens **35715**
> - **SSR**: offline **6.4 GPU-h**, latency **8.1 s**, completion tokens **14114**
>
> For SSR, the online latency further decomposes into: retrieval **0.11 s**, graph query **0.05 s**, guided generation **7.94 s**. Thus, SSR has nonzero offline cost, but the **online non-generation overhead is small relative to generation**, and the per-query online burden is much lower than ToT. The slightly lower completion-token count of SSR than DS is due to guidance reducing exploratory/repetitive reasoning and shortening the final trace.
>
> **3) ToT comparison: calibrated efficiency/accuracy claim.**
> We agree that ToT and SSR have different compute profiles, so universal “efficiency superiority” is not the right framing. ToT is training-free but incurs much higher online branching cost; SSR shifts part of the cost offline and then performs one guided generation without tree expansion.
>
> To make the offline stage explicit: SSR does **not** fine-tune the base reasoning LLM. The trainable SSR modules are lightweight (about **0.65M** trainable parameters in total, roughly **0.005%** of a 14B base model), and there is **no gradient update on the base LLM itself**. We agree, however, that this offline cost is nonzero and should be counted explicitly as a **one-time deployment-build cost**.
>
> **4) Eq. (4)–(5) notation ambiguity: clearer stage separation + ablation.**
> We agree the variable reuse can confuse readers. In the revision we will separate notation explicitly:
> - \(y\in\{0,1\}\): trial-level execution outcome
> - \(\tilde{u}\): aggregated/posterior executability target
> - \(\hat{u}_\theta\): predictor output
>
> We will also make the two stages explicit:
> (1) **contrastive representation learning** for feature shaping;
> (2) **supervised executability prediction** for final ranking.
>
> To clarify that these stages are distinct but complementary, we compare a variant **without contrastive representation learning** against full SSR.
>
> *Results*
> - **w/o contrastive representation learning**: HM-RB **68.60**, AIME25 **70.33**, Apex **13.06**
> - **Full SSR**: HM-RB **70.20**, AIME25 **74.67**, Apex **14.69**
>
> In the revised manuscript, we will (i) explicitly state \(\alpha=\beta=1\) and include prior sensitivity results, (ii) add complete cost tables with both offline and online components, (iii) recalibrate SSR-vs-ToT wording to emphasize deployment trade-offs, and (iv) clean up Eq. (4)–(5) notation and stage descriptions. We believe these changes directly address the reviewer’s concerns.

---

> > ### Author Rebuttal · Reviewer_UMWQ · 2026-04-03
> >
> > Thanks for your response. My concerns have been solved and I will raise my score.

---

> > > ### Author Response · Authors · 2026-04-03
> > >
> > > Thank you very much for your follow-up and for recognizing our efforts in the rebuttal. We are very glad that the additional analyses and clarifications addressed your concerns. We sincerely appreciate your positive assessment and your willingness to raise the score. In the final version, we will carefully incorporate all promised revisions to further improve the clarity and rigor of the paper.

---

### Decision · Program_Chairs · 2026-04-30

**Decision:**

Accept (regular)

**Comment:**

This paper introduces the distinction between strategy usage and strategy executability in mathematical reasoning guidance, showing that strategies appearing in successful solutions are not necessarily effective when provided as guidance to a target model. Based on this diagnosis, the authors propose Selective Strategy Retrieval (SSR), a test-time framework that ranks and combines strategies by empirically estimated executability. The paper received scores of 5 (Accept), 4 (Weak Accept), 4 (Weak Accept), and 3 (Weak Reject). Reviewer JRZX maintained a weak reject. Having reviewed the exchange, I find Reviewer JRZX's concerns have been largely addressed by the authors. The core conceptual contribution of the paper is well-supported and practically useful, though the technical instantiation is largely standard. I recommend weak accept.